# Methane mapping, emission quantification, and attribution in two European cities; Utrecht, NL and Hamburg, DE

Hossein Maazallahi[1,2], Julianne M. Fernandez[3], Malika Menoud[1], Daniel Zavala-Araiza[1,4],
Zachary D. Weller[5], Stefan Schwietzke[6], Joseph C. von Fischer[7], Hugo Denier van der Gon[2], and
Thomas Röckmann[1]

[1]Institute for Marine and Atmospheric research Utrecht (IMAU), Utrecht University (UU), Utrecht, The Netherlands
[2]Netherlands Organisation for Applied Scientific Research (TNO), Utrecht, The Netherlands
[3]Department of Earth Sciences, Royal Holloway University of London (RHUL), Egham, United Kingdom
[4]Environmental Defense Fund (EDF), Utrecht, The Netherlands
[5]Department of Statistics, Colorado State University (CSU), United States of America
[6]Environmental Defense Fund (EDF), Berlin, Germany
[7]Department of Biology, Colorado State University (CSU), United States of America

*Correspondence to*: Hossein Maazallahi (h.maazallahi@uu.nl)

**Abstract.** Characterizing and attributing methane ($CH_4$) emissions across varying scales is important from environmental, safety, and economic perspectives, and is essential for designing and evaluating effective mitigation strategies. Mobile real-time measurements of $CH_4$ in ambient air offer a fast and effective method to identify and quantify local $CH_4$ emissions in urban areas. We carried out extensive campaigns to measure $CH_4$ mole fractions at the street level in Utrecht, The Netherlands (2018 and 2019) and Hamburg, Germany (2018). We detected 145 leak indications (LIs, i.e., $CH_4$ enhancements of more than 10% above background levels) in Hamburg and 81 LIs in Utrecht. Measurements of the ethane-to-methane ratio ($C_2:C_1$), methane-to-carbon dioxide ratio ($CH_4:CO_2$), and $CH_4$ isotope composition ($\delta^{13}C$ and $\delta D$) show that in Hamburg about 1/3 of the LIs, and in Utrecht 2/3 of the LIs (based on a limited set of $C_2:C_1$ measurements), were of fossil fuel origin. We find that in both cities the largest emission rates in the identified LI distribution are from fossil fuel sources. In Hamburg, the lower emission rates in the identified LI distribution are often associated with biogenic characteristics, or partly combustion. Extrapolation of detected LI rates along the roads driven to the gas distribution pipes in the entire road network yields total emissions from sources that can be quantified in the street-level surveys of $440 \pm 70$ t yr$^{-1}$ from all sources in Hamburg, and $150 \pm 50$ t yr$^{-1}$ for Utrecht. In Hamburg, $C_2:C_1$, $CH_4:CO_2$, and isotope-based source attributions shows that 50 - 80 % of all emissions originate from the natural gas distribution network, in Utrecht more limited attribution indicates that 70 - 90 % of the emissions are of fossil origin. Our results confirm previous observations that a few large LIs, creating a heavy tail, are responsible for a significant proportion of fossil $CH_4$ emissions. In Utrecht, 1/3 of total emissions originated from one LI and in Hamburg >1/4 from 2 LIs. The largest leaks were located and fixed quickly by GasNetz Hamburg once the LIs were shared, but 80 % of the (smaller) LIs attributed to the fossil category could not be detected/confirmed as pipeline leaks. This issue requires further investigation.

## 1 Introduction

Methane ($CH_4$) is the second most important anthropogenic greenhouse gas (GHG) after carbon dioxide ($CO_2$) with a global warming potential of 84 compared to $CO_2$ over a 20-year time horizon (Myhre et al., 2013). The increase of $CH_4$ mole fraction from about 0.7 parts per million (ppm) or 700 parts per billion (ppb) in pre-industrial times (Etheridge et al., 1998; MacFarling Meure et al., 2006) to almost 1.8 ppm at present (Turner et al., 2019) is responsible for about 0.5 W m$^{-2}$ of the total 2.4 W m$^{-2}$ radiative forcing since 1750 (Etminan et al., 2016; Myhre et al., 2013). In addition to its direct radiative effect, $CH_4$ plays an important role in tropospheric chemistry and affects the mixing ratio of other atmospheric compounds, including direct and indirect greenhouse gases, via reaction with the hydroxyl radical (OH), the main loss process of $CH_4$ (Schmidt and

Shindell, 2003). In the stratosphere $CH_4$ is the main source of water vapor ($H_2O$) (Noël et al., 2018), which adds another aspect
to its radiative forcing. Via these interactions the radiative impact of $CH_4$ is actually higher than what can be ascribed to its
mixing ratio increase alone, and the total radiative forcing ascribed to emissions of $CH_4$ is estimated to be almost 1 W m$^{-2}$, $\approx$
60 % of that of $CO_2$ (Fig 8.17 in Myhre et al., 2013). Given this strong radiative effect, and its relatively short atmospheric
lifetime of about $9.1 \pm 0.9$ yr (Prather et al., 2012), $CH_4$ is an attractive target for short- and medium-term mitigation of global
climate change as mitigation will yield rapid reduction in warming rates.

$CH_4$ emissions originate from a wide variety of natural and anthropogenic sources, for example emissions from

natural wetlands, agriculture (e.g. ruminants or rice agriculture), waste decomposition, or emissions (intended and non-
intended) from oil and gas activities that are associated with production, transport, processing, distribution, and end-use of
fossil fuel sector (Heilig, 1994). Fugitive unintended and operation-related emissions occur across the entire oil and natural
gas supply chain. In the past decade, numerous large studies have provided better estimates of the emissions from extended
oil and gas production basins (Allen et al., 2013; Karion et al., 2013; Omara et al., 2016; Zavala-Araiza et al., 2015; Lyon et
al., 2015), the gathering and processing phase (Mitchell et al., 2015), and transmission and storage (Zimmerle et al., 2015;
Lyon et al., 2016) in the United States (US). A recent synthesis concludes that the national emission inventory of the US
Environmental Protection Agency (EPA) underestimated supply chain emissions by as much as 60 % (Alvarez et al., 2018).
McKain et al. (2015) discussed how inventories may underestimate the total $CH_4$ emission for cities. Also, an analysis of
global isotopic composition data suggests that fossil related emissions may be 60 % higher than what has been previously
estimated (Schwietzke et al., 2016). A strong underestimate of fossil fuel related emissions of $CH_4$ was also implied by analysis
of $\delta^{14}$C-$CH_4$ in pre-industrial air (Hmiel et al., 2020). These emissions do not only have adverse effects on climate, but also
represent an economic loss (Xu and Jiang, 2017) and a potential safety hazard (West et al., 2006). While $CH_4$ is the main
component in natural gas distribution networks (NGDNs), composition of natural gas varies from one country or region to
another. In Europe the national authorities provide specifications on components of natural gas in the distribution network
(Table 8 in UNI MISKOLC and ETE, 2008).

Regarding $CH_4$ emissions from NGDNs, a number of intensive $CH_4$ surveys with novel mobile high precision laser-

based gas analyzers in US cities have recently revealed the widespread presence of leak indications (LIs: $CH_4$ enhancements
of more than 10 % above background level) with a wide range of magnitudes (Weller et al., 2020; Weller et al., 2018; von
Fischer et al., 2017; Chamberlain et al., 2016; Hopkins et al., 2016; Jackson et al., 2014; Phillips et al., 2013). The number and
severity of natural gas leaks appears to depend on pipeline material and age, local environmental conditions, pipeline
maintenance and replacement programs (von Fischer et al., 2017; Gallagher et al., 2015; Hendrick et al., 2016). For example,
NGDNs in older cities with a larger fraction of cast iron or bare steel pipes showed more frequent leaks than NGDNs that use
the newer plastic pipes. The data on $CH_4$ leak indications from distribution systems in cities have provided valuable data for
emission reduction in the US cities which allows local distribution companies (LDCs) who are in charge of NGDN to quickly
fix leaks and allocate resources efficiently (Weller et al., 2018, von Fischer et al., 2017, Lamb et al., 2016; McKain et al.,

2015).

Urban European cities $CH_4$ emissions are not well known, which requires carrying out extensive campaigns to collect

required observation data. Few studies have estimated urban $CH_4$ fluxes using eddy covariance measurements (Gioli et al.,
2012; Helfter et al., 2016), airborne mass balance approaches (O'Shea et al., 2014) and the Radon-222 flux and mixing layer
height techniques (Zimnoch et al., 2019). Gioli et al. (2012) showed that about 85 % of methane emissions in Florence, Italy
originated from natural gas leaks. Helfter et al. (2016) estimated $CH_4$ emissions of $72 \pm 3$ t km$^{-2}$ yr$^{-1}$ in London, UK mainly
from sewer sesytem and NGDNs leaks, which is twice as much as reported in the London Atmospheric Emissions Inventory.
O'Shea et al. (2014) also showed that $CH_4$ emissions in greater London is about 3.4 times larger than the report from UK
National Atmospheric Emission Inventory. Zimnoch et al. (2019) estimated $CH_4$ emissions of $(6.2 \pm 0.4) \times 10^6$ m$^3$ year$^{-1}$ for
Krawko, Poland, based on data for the period of 2005 to 2008 and concluded that leaks from NGDNs are the main emission
source in Krawko, based on carbon isotopic signature of $CH_4$. Chen et al. (2020) also showed that incomplete combustion or
loss from temporarily installed natural gas appliances during big festivals can be the major source of $CH_4$ emissions from such
events, while these emissions have not been included in inventory reports for urban emissions.
Here we present the result of mobile in-situ measurements at street level for whole-city surveys in two European
cities, Utrecht in the Netherlands (NL) and Hamburg in Germany (DE). In this study, we quantified LIs emissions using an
empirical equation from Weller et al. (2019), which was designed based on controlled release experiments from von Fischer
et al. (2017), to quantify ground-level emissions locations in urban area such as leaks from NGDN. In addition to finding and
categorizing the $CH_4$ enhancements (in a similar manner as done for the US cities in order to facilitate comparability), we
made three additional measurements to better facilitate source attribution: the concomitant emission of ethane ($C_2H_6$) and $CO_2$,
and the carbon and hydrogen isotopic composition of the $CH_4$. These tracers allow an empirically based source attribution for
LIs. In addition to emission quantifications of LIs across the urban areas in these two cities, we also quantified $CH_4$ emissions
from some of facilities within the municipal boundary of Utrecht and Hamburg using Gaussian plume dispersion model
(GPDM).
**2 Materials and methods**
**2.1 Data collection and instrumentation**
**2.1.1 Mobile measurements for attribution and quantification**
Mobile atmospheric measurements at street level were conducted using two Cavity Ring-Down Spectroscopy (CRDS)
analyzers (Picarro Inc. model G2301 and G4302) which were installed on the back seat of a 2012 Volkswagen Transporter,
(see supplementary information (SI), Sect. S.1.1, Figure S1). The model G2301 instrument provides atmospheric mole fraction
measurements of $CO_2$, $CH_4$ and $H_2O$, each of them with an integration time of about 1 s., which results in a data frequency of
$\approx 0.3$ Hz for each species. The reproducibility for $CH_4$ measurements was $\approx 1$ ppb for 1 s integration time. The G2301
instrument was powered by a 12 V car battery via a DC-to-AC converter. The flow rate was $\approx 187$ ml min$^{-1}$. Given the volume
and pressure of the measurement cell (volume = 50 ml and pressure $\approx 190$ mbar) the cell is flushed approximately every 3 s,
so observed enhancements are considerably smoothed out. The factory settings for $CH_4$ and $CO_2$ were used for the water
correction.
The G4302 instrument is a mobile analyzer that provides atmospheric mole fraction measurements of $C_2H_6$, $CH_4$, and
$H_2O$. The flow rate is 2.2 L min$^{-1}$ and the volume of the cell is 35 ml (operated at 600 mb, thus 21 ml STP) so the cell is flushed
in 0.01 s, which means that mixing is insignificant given the 1 s measurement frequency of the G4302. The additional
measurement of $C_2H_6$ is useful for source attribution since natural gas almost always contains a significant fraction of $C_2H_6$,
whereas microbial sources generally do not emit $C_2H_6$ (Yacovitch et al., 2014). The G4302 runs on a built-in battery which
lasts for $\approx 6$ h. The instrument can be operated in two modes at $\approx 1$ Hz frequency for each species: the $CH_4$-only mode and the
$CH_4$ - $C_2H_6$ mode. In the $CH_4$-only mode the instrument has a reproducibility of $\approx 10$ ppb for $CH_4$. The factory settings for $CH_4$
and $C_2H_6$ were used for the water correction. In the $CH_4$ - $C_2H_6$ mode the reproducibility is about 100 ppb for $CH_4$ and 15 ppb
for $C_2H_6$. For Utrecht surveys (see SI, Sect. S.1.2, Figure S2a), the G4302 was not yet available for the initial surveys in 2018,
but it was added for the later re-visits (see SI, Sect. S.1.2, Table S1). For Hamburg (see SI, Sect. S.1.2, Figure S2b), both
instruments operated during the entire intensive 3-week measurement campaign in Oct/Nov 2018 (see SI, Sect. S.1.2, Table
S2). The time delay from the inlet to the instruments was measured and accounted for in the data processing procedure. The
Coordinated Universal Time (UTC) time shifts between the Global Positioning System (GPS) and the two Picarro instruments
were corrected for each instrument in addition to the inlet delay (see SI, Sect. S.1.2, Table S1 and Table S2). The clocks on
the Picarro instruments were set to UTC but showed drift over the period of the campaigns. We recorded the drifts for each
day's survey and corrected to UTC time. The data were also corrected for the delay between air at the inlet and the signal in
the $CH_4$ analyzers. This delay was determined by exposing the inlet to three small $CH_4$ pulses from exhaled breath, ranging
from 5-30 seconds, depending on the instrument and tubing length. We averaged the three attempts to determine the delay for
each instrument and used the delays for each instrument. Individual attempts were 1 to 2 s different from each other. For the
G4302 the delay was generally about 5 s and for the G2301 it was about 30 s; the difference is mainly due to the different flow
rates. The recorded $CH_4$ mole fractions were projected back along the driving track according to this delay.
One-quarter inch Teflon tubing was used to pull in air either from the front bumper (0.5 m above ground level) to the
G2301 or from the rooftop (2 m above ground level) to the G4302. To avoid dust into the inlets for both instruments, Acrodisc®
syringe filter, 0.2 μm was used for G2301 and Parker Balston 9933-05-DQ was used for G4302. The G2301 was used for
quantification and attribution purposes and the G4302 mainly for attribution. After data quality check, a comparison between
the two instruments during simultaneous measurements showed that all LIs were detectable by both instruments despite
difference in inlet height (see SI, Sect. S.1.3, Figure S3). A comparison between the two instruments during simultaneous
measurements showed that all LIs were detected by both instruments despite difference in instrument characteristics and inlet
height. In the majority of cases $CH_4$ enhancements for each LI from both instruments were similar to each other. We note that
there is likely a compensation of differences from two opposing effects between the two measurement systems. The inlet of
the G2301 was at the bumper, thus closer to the surface sources, but the rather low flow rate and measurement rate of the
instrument lead to some smoothing of the signal in the cavity. Because of the high gas flow rate, signal smoothing is much
reduced for the G4302, but the inlet was on top of the car, thus further away from the surface sources (see Table S3 in SI, Sect.
S.1.3). The vehicle locations were registered using a GPS system that recorded the precise driving track during each survey.

### 2.1.2 Target cities: Utrecht and Hamburg

Utrecht is the 4[th] largest city in the Netherlands with population of approximately 0.35 million inhabitants within an
area of roughly 100 km². It is located close to the center of the Netherlands and is an important infrastructural hub in the
country. The Utrecht city area that we target in this study is well constrained by a ring of highways around the city (A27, A12,
A2, and N230) with inhabitants of approximately 0.28 million living within this ring on roughly 45 km² of land. Figure S2a
(see SI, Sect. S.1.2) shows the streets that were driven in Utrecht and Figure 1a shows the street coverage over four street
categories (level 1, 2, 3, residential, and unclassified) obtained from the Open Street Map (OSM; www.openstreetmap.org).
Table S4 (see SI, Sect. S.1.5) provides information on road coverage based on different street categories. The hierarchy of
OSM road classes is based on the importance of roads in connecting parts of the national infrastructure. Level 1 roads are
primarily larger roads connecting cities, level 2 roads are the second most important roads and part of a greater network to
connect smaller towns, level 3 roads have tertiary importance level and connect smaller settlements and districts. Residential
roads are roads which connect houses and unclassified roads have the lowest importance of interconnecting infrastructure.
Moreover, several transects were also made to measure the atmospheric mole fraction of $CH_4$ from the road next to the waste
water treatment plant (WWTP) in Utrecht – a potentially larger single source of $CH_4$ emissions in the city (see SI, Sect. S.1.6,
Table S5).
Hamburg is the 2[nd] largest city in Germany (about 1.9 million inhabitants, 760 km² area) and hosts one of the largest
harbors in Europe. The study area in Hamburg is North of the Elbe river (Figure 1b) with ≈1.4 million inhabitants on about
400 km² land. Figure S2b (see SI, Sect. S.1.2) shows the streets that were covered in Hamburg and Figure 1b shows the street
coverage categorized in the four categories of OSM. More information on road coverage based on OSM street categories are
provided in Table S4 (see SI, Sect. S.1.5). The local distribution companies (LDCs) in Utrecht (STEDIN
([https://www.stedin.net/](https://www.stedin.net/))) and Hamburg (GasNetz Hamburg ([https://www.gasnetz-hamburg.de](https://www.gasnetz-hamburg.de))) confirmed that full pipeline
coverages are available beneath all streets. Therefore, the length of roads in the study area of Utrecht and Hamburg are
representatives of NGDNs length. The Hamburg harbor area hosts several large industrial facilities that are related to the
midstream / downstream oil and gas sector including refineries and storage tanks. An oil production site (oil well, separator
and storage tanks) at Allermöhe (in Hamburg-Bergedorf) was also visited. Information from the State Authority for Mining,
Energy and Geology (LBEG, 2018) was used to locate facilities. Precise locations of the facilities surveyed are given in the
Table S6 (see SI, Sect. S.1.6). In order to separate these industrial activities from the NGDNs emissions in this study, $CH_4$
emissions from these locations were estimated, but evaluated apart from the emissions found in each city. The reported in-situ
measurement, GPS data, and boundary of study areas reported here are available on the Integrated Carbon Observation System
(ICOS) portal (Maazallahi et al., 2020b).
**2.1.3 Driving strategy**
The start/end point for each day's measurement surveys across Utrecht and Hamburg were the Institute for Marine
and Atmospheric research Utrecht (IMAU; Utrecht University) and the Meteorological Institute (MI; Hamburg University),
respectively. From these starting locations, each day's surveys targeted the different districts and neighborhoods of the cities
(see SI, Sect. S.1.2, Table S1 and Table S2). Measurement time periods and survey areas were chosen to select favorable traffic
and weather conditions and to avoid large events (e.g., construction; see SI, Sect. S.1.5, Figure S4), which normally took place
between 10 - 18 LT. Average driving speeds on city streets were in the range of $17 \pm 7$ km h$^{-1}$ in Utrecht and $20 \pm 6$ km h$^{-1}$ in
Hamburg.
As part of our driving strategy, we revisited locations where we had observed enhanced $CH_4$ readings (see SI, Sect.
S.1.7, Figure S5). Not all recorded $CH_4$ mole fraction enhancements are necessarily the result of a stationary $CH_4$ source. For
example, they could be related to emissions from vehicles which run on compressed natural gas, or vehicles operated with
traditional fuels but with faulty catalytic converter systems. Later we will discuss how to exclude or categorize these
unintended signals (see Sect. 2.2.2 and Sect. 2.3.1). Therefore, we revisited a large number of locations (65 in Utrecht ($\approx$80
%) and 100 in Hamburg ($\approx$70 %)) where enhanced $CH_4$ had been observed in during the first survey in order to confirm the
LIs. In contrast to the measurements carried out in many cities in the United States (US) (von Fischer et al., 2017), our
measurements were not carried out using Google Street View cars, but with a vehicle from the Institute for Marine and
Atmospheric research Utrecht (IMAU), Utrecht University (see SI, Sect. S.1.1, Figure S1). Due to time and budget restrictions,
it was not possible to cover each street at least twice, as done for the US cities. After evaluation of the untargeted first surveys
that covered each street at least once, targeted surveys were carried out for verification of observed LIs and for collection of
air samples at locations with high $CH_4$ enhancements. The rationale behind this measurement strategy is that if an enhancement
was not recorded during the first survey, it obviously cannot be verified in the second survey. The implications of the difference
in the measurement strategy will be discussed in the Results and Discussion sections below.
In total, approximately 1,300 km of roads were driven during Utrecht surveys and about 2,500 km during the Hamburg
campaign. In Utrecht, some re-visits were carried out several months to a year after the initial surveys in order to check on the
persistence of the LIs. In Hamburg, revisits were also performed within the 4-week intensive measurement period. Further
details about the driving logistics are provided in the SI (Sect. S.1.6, Table S1 and Table S2). It is possible that pipeline leaks
that were detected during the initial survey were repaired before the revisit, and the chance of this occurring increases as the
time interval between visits gets longer.

### 2.1.4 Air sample collection for attribution

In addition to the mobile measurement of $C_2H_6$ and $CO_2$ for LIs attributions purposes, samples for lab isotope analysis of $\delta^{13}C$-$CH_4$ and $\delta^2H$-$CH_4$ (hereinafter $\delta^{13}C$ and $\delta D$ respectively) were collected during the revisits at locations that had displayed high $CH_4$ enhancements during the first surveys. Depending on the accessibility and traffic, samples were either taken inside the car (see SI, Sect. S.1.8, Figure S6a) using a tubing from the bumper inlet, or outside the car on foot using the readings from the G4302 to find the best location within the plume (see SI, Sect. S.1.8, Figure S6b). All the samples taken in the North Elbe study area and from most of the facilities were collected when the car was parked, but the samples inside the New Elbe tunnel and close to some facilities where there was no possibility to park were taken in motion while we were within the plume. The sampling locations across the North Elbe study area of Hamburg were determined based the untargeted surveys, and the confirmation during revisits. The $C_2H_6$ information was not used in the selection of sampling locations in order to avoid biased sampling. Sampling locations from the facilities were determined based on wind direction, traffic, and types of different activities. Samples for isotope analysis were collected in non-transparent aluminum-coated Tedlar Supelco, Seupel™ Inert SCV Gas Sampling Bag (2 L) and SKC, Standard FlexFoil® Air Sample Bags (3 L) using a 12 V pump and 1/4-inch Teflon tubing which pumps air with flow rate of ≈0.25 L min$^{-1}$. In total, 103 bag samples were collected at 24 locations in Hamburg, 14 of them in the city area North of the Elbe river and 10 at larger facilities. Usually, three individual samples were collected at each source location, plus several background air samples on each sampling day. This sampling scheme generally results in a range of mole fractions that allow source identification using a Keeling plot analysis (Keeling, 1958, 1961). Fossil $CH_4$ sources in the study areas of this paper (inside the ring for Utrecht and north Elbe in Hamburg) refers to emissions originating from natural gas leaks.

### 2.1.5 Meteorological Data

Meteorological information reflecting the large scale wind conditions during the campaigns were obtained from measurements at the Cabauw tower (51.970263° N, 4.926267° E) operated by Koninklijk Nederlands Meteorologisch Instituut (KNMI) (Van Ulden and Wieringa, 1996) for Utrecht and Billwerder tower (53.5192° N, 10.1029° E) operated by the MI at Hamburg University (Brümmer et al., 2012) for Hamburg. The wind direction and wind speed data from the masts were used for planning the surveys. Pressure and temperature measurements were used to convert volume to mass fluxes for $CH_4$. We also used information from the towers for the GPDM calculations of the emission rates from larger facilities, because the local wind measurements from the 2-D anemometer were not logged continuously due to failure in logging setup of the measurements. In Utrecht, the Cabauw tower is located about 20 km from the WWTP. In Hamburg Billwerder tower is about 18 km from the Soil and Compost company and about 8 km from oil production facilities. Uncertainties over the wind data will be described later.

### 2.2 Emission quantification

### 2.2.1 Data preparation and background extraction of mobile measurements

The first step of the evaluation procedure is quality control of the data from both $CH_4$ analyzers and the GPS records. Periods of instrument malfunction and unintended signals based on notes written during each day's measurements were removed from the raw data. Extraction of the LIs from in-situ measurements requires estimation of the background levels (see SI, Sect. S.2.1, Figure S7). We estimated $CH_4$ background as the median value of ± 2.5 min of measurements around each individual point as suggested in Weller et al. (2019). For estimating the $CO_2$ background level we used the 5$^{th}$ percentile of ± 2.5 min of measurements around each individual point (Brantley et al., 2014; Bukowiecki et al., 2002). The background determination method for $CH_4$ was selected from Weller et al. (2019) to follow the emission quantification algorithm for the

urban studies, and while this algorithm doesn't include background extraction for $CO_2$, we chose commonly adopted method
of background determination for this component. These background signals were subtracted from the measurement time series
to calculate the $CH_4$ and $CO_2$ enhancements. For $C_2H_6$, the background was considered zero as it is normally present at a very
low mole fraction; between ~0.4-2.5 ppb (Helmig et al., 2016), and is lower than the G4302 detection limit.

### 2.2.2 Quantification of methane emissions from leak indications

We wrote an automated MATLAB® script (available on GitHub from Maazallahi et al. (2020a)) based on the
approach initially introduced in von Fischer et al. (2017), and improved in Weller et al. (2019). This algorithm was designed
to quantify $CH_4$ emissions from ground-level emission release locations within 5-40 m from the measurement (von Fischer et
al., 2017), such as pipeline leaks and has been demonstrated that the algorithm adequately estimates the majority of those
emissions from a city (Weller et al., 2018). Using the same algorithm also ensures that results are comparable between
European and US cities. The individual steps will be described below. Mapping and spatial analysis were conducted using
Google Earth and ESRI ArcMap software. A flow diagram of the evaluation procedure is provided in the SI (Sect. S.2.2, Figure
S8).
Following the algorithm from von Fischer et al. (2017), measurements at speeds above 70 km h$^{-1}$ were excluded, as
the data from the controlled release experiments (von Fischer et al., 2017) were not reliable at high speed (Weller et al., 2019).
We also excluded measurements during periods of zero speed (stationary vehicle) to avoid unintended signals coming from
other cars running on compressed natural gas when the measurement car was stopped in traffic. In order to merge the sharp 1
Hz-frequency records of the GPS with the ≈ 0.3 Hz data from the G2301 analyzer, the $CH_4$ mole fractions were linearly
interpolated to the GPS times.
Weller et al., (2019) established an empirical equation to convert LIs observed with a Picarro G2301 in a moving
vehicle in urban environments into emission rates based on a large number of controlled release experiments in various
environments (Eq. (1)).
$\text{Ln}(C) = -0.988 + 0.817 * \text{Ln}(Q)$            (1)
In this equation, C represents $CH_4$ enhancements above the background in ppm and Q is the emission rate in L min$^-$
$^1$. Weller et al., (2019) used controlled releases to demonstrate that the magnitude of the observed methane enhancement is
related to the emission rate and carefully characterized the limitations and associated errors of this equation. We used Eq. (1)
to convert $CH_4$ enhancements encountered during our measurements in Utrecht and Hamburg to emission rates, and we use
these estimates to categorize LIs into three classes: high (emission rate > 40 L min$^{-1}$), medium (emission rate 6− 40 L min$^{-1}$)
and low (emission rate 0.5 - 6 L min$^{-1}$), following the categories from von Fischer et al. (2017) (Table 1).
The spatial extent of individual LIs was estimated as the distance between the location where the $CH_4$ mole fraction
exceeded the background by more than 10 % (≈ 0.200 ppm; as used in von Fischer et al. (2017) and Weller et al. (2019)) to
the location where it fell below this threshold level again. LIs which stay above the threshold for more than 160 m were
excluded in the automated evaluation because we suspect that such extended enhancements are most likely not related to leaks
from the NGDN (von Fischer et al., 2017).
In a continuous measurement survey on a single day, consecutive $CH_4$ enhancements above background observed
within 5 seconds were aggregated and the location of the emission source was estimated based on the weighted averaging of
coordinates (Eq. (2)). Decimal degree coordinates were converted to Cartesian coordinates (see SI, Sect. S.2.3, Figure S9)
relative to local references (see SI, Sect. S.2.3, Table S7). In Utrecht, the Cathedral tower (Domtoren) and in Hamburg the St.
Nicholas' Church were selected as local geographic datums. LIs observed on different days at similar locations were clustered
and interpreted as one point source when circles of 30 m radius around the centre locations overlapped, similar to Weller et
al., (2019). The enhancement of the cluster was assigned the maximum observed mole fraction and located as the weighted
average of the geographical coordinates of the LIs within that cluster (Eq. (2) from Weller et al. (2019)), where $w_i$ is $CH_4$
enhancement of each LI.
$(lon, lat) = \frac{\sum_{i=1}^{n} w_i*(lon_i, lat_i)}{\sum_{i=1}^{n} w_i}$ (2)
We compared the outputs of our software to the one developed by Colorado State University (CSU) for the surveys
in US cities (von Fischer et al., 2017; Weller et al., 2019). 30 LIs were detected and no significant differences were observed
(linear fit equation y = 1.00 * x - 0.00, $R^2$ = 0.99) (see SI, Sect. S.2.4, Figure S10). As mentioned above, in our campaign-type
studies not all streets were visited twice, so this criterion was dropped from the CSU algorithm. Instead, we used explicit
source attribution by co-emitted tracers.
The emission rate per km of road covered during our measurements was then scaled up to the city scale using the
ratio of total road length within the study area boundaries derived from OSM to the length of streets covered, and converted
to a per-capita emission using the population in the study areas based on LandScan data (Bright et al., 2000). Note that in this
up-scaling practice, emission quantified from facilities were excluded.
To account for the emission uncertainty, similar to Weller et al. (2018) for the US city studies, we used a bootstrap
technique which was initially introduced in Efron (1979, 1982), as this technique is adequate in resampling of both parametric
and non-parametric problems with even non-normal distribution of observed data. Tong et al. (2012) indicated that bootstrap
resampling technique is sufficiently capable in estimating uncertainty of emissions with sample size of equal or larger than 9.
Efron and Tibshirani (1993) suggested that minimum of 1,000 iterations are adequate in bootstrap technique. In this study, we
used non-parametric bootstrap technique to account for the uncertainty of total $CH_4$ emissions from all LIs in each city with
30,000 replications. As mentioned above the algorithm is based on $CH_4$ enhancements of measurement with 5-40 m distance
from controlled release location, and can produce large uncertainty for emission quantification of individual LI (Figure 4 in
Weller et al. (2019)), but with sufficient number of sample size, the uncertainty associated with total emission quantified in an
urban area is more precise.
**2.2.3 Quantification of methane emissions from larger facilities**
Apart from the natural gas distribution network, there are larger facilities in both cities that are potential $CH_4$ sources
within the study area. Several facilities in or around the cities were visited during the mobile surveys to provide emission
estimates. We applied a standard point source GPDM (Turner, 1969) to quantify methane emissions from these larger facilities.
A flowchart describing the steps taken during quantification from facilities in given in SI (Sect. S.2.5., Figure S11). We note
that emission quantification using GPDM with data from mobile measurements is prone to large errors (factor of 3 or more )
(Yacovitch et al., 2018) especially when the measurements are carried out close to the source. In this study, we also report the
data obtained from larger facilities, since rough emission estimates from facilities can be obtained in the city surveys. Caulton
et al. (2018) discuss uncertainties of emission quantification with GPDM. Individual facilities were visited during the routine
screening measurements and during revisits for LI confirmation and air sampling.
In Utrecht, the WWTP is located in the study area and streets around this facility were passed several times during
surveys. In Hamburg, we initially performed screening measurements in the harbor area (extensive industrial activities) and
near an oil production site and then revisited these sites for further quantification and isotopic characterization. The data from
the oil production site can be fit reasonably well with a GPDM and were therefore selected for quantification, similar to studies
in a shale gas production basin in the USA (Yacovitch et al., 2015) and in the Netherlands (Yacovitch et al., 2018).
$C(x,y,z) = \frac{Q}{2*\pi*u*\sigma_y*\sigma_z} * \{ \exp(\frac{-(z-z_{source})^2}{2*\sigma_z^2}) + \exp(\frac{-(z+z_{source})^2}{2*\sigma_z^2}) \} * \exp(\frac{-y^2}{2*\sigma_y^2}) \}$

(3)

In Eq. (3), C is the $CH_4$ enhancement converted to the unit of $g/m^3$ at cartesian coordinates x, y, and z relative to the
source ($[x\ y\ z]_{source}$ = 0), x is the distance of the plume from the source aligned with the wind direction, y is the horizontal axis
perpendicular to the wind direction, z is the vertical axis. Q is emission rate in g s$^{-1}$, u (m s$^{-1}$) is the wind speed along the x-
axis, and $\sigma_y$ and $\sigma_z$ are the horizontal and vertical plume dispersion parameters (described below), respectively.

Determination of an effective release location is a challenge for the larger facilities. Effective emission locations for
each facility were estimated based on wind direction measurements and the locations of maximum $CH_4$ enhancements. The
facilities were generally visited multiple times under different wind conditions. The locations of the maximum $CH_4$
enhancements were then projected against the ambient wind, and the intersection point of these projections during different
wind conditions was defined as effective emission location of the facility. At least two measurement transects with different
wind direction were used to estimate the effective location of the source. If wind directions, road accessibility or the shape of
plumes were not sufficient to indicate the effective source location, the geographical coordinates of centroids of the possible
sources using Google Earth imageries and field observations were used to determine the effective emission location. For the
WWTP in Utrecht we also contacted the operator and asked for the location of sludge treatment as it is the major source of
$CH_4$ emissions (Paredes et al., 2019; Schaum et al., 2015).

Neumann and Halbritter (1980) showed that the main parameters in sensitivity analysis of GPDM are the wind speed
and source emission height in close distance and the influence of emission height become less further downwind compared to
the mixing layer height. In this study, the heights of emission sources were low (<10m) and estimated during surveys and/or
using Google Earth imageries, and considering that such a larger measurement distance from the facilities, the main sources
of uncertainty of the emission estimates for the WWTP and Compost and Soil company are most likely the mean wind speed
and for the upstream facilities in Hamburg the major sources of uncertainties can be the mean wind speed and emission height.
We considered 0-4 m source height for the WWTP in Utrecht, and for the upstream facilities in Hamburg we considered 0-5
m emission height for the Compost and Soil site, 0-2 m for the separator, 0-10 m for the storage tank, and 0-1 m for the oil
extraction well-head. We used 1 m interval for each of these height ranges to quantify emissions in GPDM.

Cross wind horizontal dispersions $\sigma_y$ were estimated from the measured plumes by fitting a Gaussian curve to the
individual plumes from each set during each day's survey. A set of plumes is defined as a back to back transects during a
period of time downwind each facility on different days. Later average emissions from all sets of plumes were used to report
$CH_4$ emission for each of the facilities. A suitable Pasquill–Gifford stability class was then determined by selecting a pair of
parameters (Table 1-1 in EPA, 1995) that matches best and give the closest number to the with the fitted value of $\sigma_y$. Vertical
dispersions $\sigma_z$ were then estimated using the identified Pasquill–Gifford stability class in the first step, using the distances to
the source locations (Table 1-2 in EPA, 1995). Uncertainties due to these estimates will be discussed below. Mass emission
rates were calculated using the metric volume of $CH_4$ at 1 bar of atmospheric pressure (0.715 kg m$^{-3}$ at 0 °C and 0.666 kg m$^{-3}$
at 20 °C, P. 1.124 in IPCC, 1996), and linear interpolation was used for temperatures in between.

Due to technical issues, local wind data were not logged continuously and thus we used wind data from two towers
which are 8 to 20 km away from the facilities we focused for emission quantifications. These distances introduce extra
uncertainties in analyzing the emissions using GPDM mainly on the wind speed. By comparing some of the local high-quality
wind data to data from the towers, we estimated that the local wind speed is within the range of ± 30 % of the collected tower
data. This range was adopted to estimate the wind speed for emission quantifications for the set of plumes measured downwind
of the facilities. The wind directions were aligned at local scale of each facility based on the locations of sources and locations
of maxima of average $CH_4$ enhancements from a set of transects in each day's survey and we considered ± 5° uncertainty in
wind direction for the GPDM quantification.

## 2.3 Emission attribution

### 2.3.1 Mobile $C_2H_6$ and $CO_2$ measurements

During the Utrecht campaign, the overall mole fraction of $CH_4$ and $C_2H_6$ in the NGDN was ≈ 80 % and ≈ 3.9 % (STEDIN, personal communication) and in Hamburg the mole fraction of $CH_4$ and $C_2H_6$ in the NGDN was about ≈ 95 % and ≈ 3.4 % (GasNetz Hamburg, personal communication) respectively. This ratio can vary depending on the mixture of gas compositions from different suppliers, but should meet the standards on the gas compositions in the Netherlands (65 – 96 mol-% for $CH_4$ and 0.2 – 11 mol-% for $C_2H_6$ (ACM, 2018)) and in Germany (83.64 – 96.96 mol-% for $CH_4$ and 1.06 – 6.93 mol-% for $C_2H_6$ (DVGW, 2013)). Compressed natural gas vehicles can be mobile $CH_4$ emission sources ( E. K. Nam et al., 2004; Curran et al., 2014; Naus et al., 2018; Popa et al., 2014) and in this study we also observed $CH_4$ signals from vehicles. For example, the point to point $C_2H_6$:$CH_4$ ratio ($C_2$:$C_1$) calculated from road measurements of a car exhaust shown in Figure S12 (see SI, Sect. S.2.6) is 14.2 ± 7.1 %. During the campaigns in Utrecht and Hamburg the $C_2$:$C_1$ of NGDNs was less than 10 % and in our study, we removed all the locations where the $C_2$:$C_1$ ratio was greater than 10 %. $CH_4$ emissions from combustion processes are always accompanied by large emissions of $CO_2$ and can therefore be identified based on the low $CH_4$:$CO_2$ emission ratio. In this study, LIs with $CH_4$:$CO_2$ ratio between 0.02 and 20 with $R^2$ greater than 0.8 were attributed to combustion.

### 2.3.2 Lab isotopic analysis of $\delta^{13}C$ and $\delta D$

After sample collections, the bag samples were returned to the IMAU for analysis of both $\delta^{13}C$ and $\delta D$ (Brass and Röckmann, 2010) and some samples were analyzed at the Greenhouse Gas Laboratory (GGL) in the department of Earth Sciences, Royal Holloway University of London (RHUL) for $\delta^{13}C$ (Fisher et al., 2006) (see SI, Sect. S.2.7, Figure S13).

At the IMAU, we used isotope ratio mass spectrometry (IRMS) instrument of ThermoFinnigan MAT DeltaPlus XL (Thermo Fisher Scientific Inc., Germany). We used a reference cylinder calibrated against Vienna Pee Dee Belmnite (V-PDB) for $\delta^{13}C$ and Vienna Standard Mean Ocean Water (V-SMOW) for $\delta D$ at the at the Max Planck Institute for Biogeochemistry (MPI-BGC), Jena, Germany (Sperlich et al., 2016). The cylinder contained $CH_4$ mole fraction of 1975.5 ± 6.3 ppb, $\delta^{13}C$ = - 48.14 ± 0.07 ‰ vs V-PDB and $\delta D$ = -90.81 ± 2.7 ‰ vs V-SMOW. The samples were pumped through a magnesium perchlorate ($Mg(ClO_4)_2$) dryer before the $CH_4$ extraction steps. Each sample was measured at least 2 times (up to four times) for each isotope. Every other sample, the reference gas was also measured 3 times for $\delta^{13}C$ and $\delta D$. Each measurement, from the $CH_4$ extraction to the mass spectrometer, took ≈ 30 minutes.

At the GGL, Flex foil SKC bag samples were each analyzed for methane mole fractions and $\delta^{13}C$. Methane mole fractions were determined using a Picarro G1301 CRDS, which measured every 5 seconds for 2 minutes resulting in a precision ± 0.3 ppb (Lowry et al., 2020; France et al., 2016; Zazzeri et al., 2015). Each sample was then measured for stable isotopes ($\delta^{13}C$-$CH_4$) using an Elementar Trace gas and continuous-flow gas chromatography isotope ratio mass spectrometry (CF-GC-IRMS) system (Fisher et al., 2006), which has an average repeatability of ± 0.05 ‰. $CH_4$ extraction was preceded by drying process using $Mg(ClO_4)_2$. Each sample was measured 3 times for $\delta^{13}C$-$CH_4$, where the duration of each analysis was ≈ 20 minutes. Both instruments are calibrated weekly to the WMO X2004A methane scale using air filled cylinders that were measured by the National Oceanic and Atmospheric Administration (NOAA), and cylinders that were calibrated against the NOAA scale by the MPI-BGC (France et al., 2016; Lowry et al., 2020).

The analytical systems for isotope analysis have been described, used and/or compared in several previous publications (Fisher et al., 2011; Röckmann et al., 2016; Umezawa et al., 2018; Zazzeri et al., 2015). Measurement uncertainties in $\delta^{13}C$ and $\delta D$ are 0.05-0.1 ‰ and 2-5 ‰ respectively.

After the LIs were analyzed and quantified, the measurements of $C_2H_6$, $CO_2$, and isotopic composition from the air
samples were used for source attribution. We characterize the observed LIs as of fossil origin when they had a concomitant
$C_2H_6$ signal between 1 % and 10 % of the $CH_4$ enhancements and when the isotopic composition was in the range -50 to -40
‰ for $\delta^{13}C$ and -150 to -200 ‰ for $\delta D$. A LI was characterized as microbial when there was no $C_2H_6$ signal (<1 % of the $CH_4$
enhancements larger than 500 ppb), $\delta^{13}C$ was between -55 ‰ and -70 ‰ and $\delta D$ was between -260 and -360 ‰ (Figure 7 in
Röckmann et al., 2016). LIs with enhancements of $CH_4$ lower than 500 ppb and no $C_2H_6$ signals were categorized as
unclassified. LIs with no $C_2H_6$ signals, no significant $CH_4:CO_2$ ratio, and no information on $\delta^{13}C$ and $\delta D$ were also categorized
as unclassified. The source signatures for each sampling location were determined by a Keeling plot analysis of the three
samples collected in the plumes and a background sample taken on the same day.
**3 Results**
**3.1 Quantification of $CH_4$ emissions across Utrecht and Hamburg**
Table 2 summarizes the main results from the surveys in Hamburg and Utrecht. The amount of km of roads covered
in Hamburg is roughly a factor of 2 larger than in Utrecht, and also the number of detected LIs is roughly a factor of 2 larger,
for all three categories. This shows that the overall density of LIs (km covered per LI) in both cities is not very different.
Specifically, a LI is observed every 5.6 km in Utrecht and every 8.4 km in Hamburg. While not all streets were visited twice
in both cities (see SI, Sect. S.1.5, Table S4) 80 % of LIs in Utrecht and 69 % of LIs in Hamburg were revisited which account
for 91 % and 86 % of emissions respectively in the study areas. During revisits, 60 % of $CH_4$ emissions in Utrecht and 46 %
of emissions in Hamburg were confirmed. In both cities, all LIs in the high emission category were re-observed. In some cases,
re-visits were carried out several months after first detection, and the LIs were still confirmed (e.g. see SI, Sect. S.1.7, Figure
S5).
The distribution of $CH_4$ LIs across the cities of Utrecht and Hamburg is shown in Figure 2. As shown in Table 2, a
total of 145 significant LIs were detected in Hamburg and 81 in Utrecht; these LIs cover all three LI categories. Two LIs in
Hamburg and one LI in Utrecht fall in the high (red) emission category; the highest LI detected in Utrecht and Hamburg
corresponded to emission rates of $\approx 100$ L min$^{-1}$ and $\approx 70$ L min$^{-1}$, respectively. Noted that estimates for individual leaks with
the Weller et al. (2019) algorithm can have large error, thus these results are indicative of large leaks, but the precise emission
strength is very uncertain. Six LIs in Utrecht and 16 LIs in Hamburg fall in the middle (orange) emission category, and 127
LIs in Hamburg and 74 LIs in Utrecht fall in the low (yellow) emission category. The distribution of emissions over the three
categories is also similar between the two cities, with roughly one third of the emissions originating from each category (Figure
2), but the number of LIs in each category is different. The contribution of LIs in the high emission category is about a third
of the total observed emissions (35 % in Utrecht is (1 LI) and in 30 % in Hamburg (2 LIs)).
$CH_4$ emitting locations were categorized based on the roads where the LIs were observed (Figure 1, Figure 2, Figure
3, and Table S8 in SI, Sect. S.3.1). Average emission rates per LI as derived from equation (1) are similar for the two cities
with 3.6 L min$^{-1}$ LI$^{-1}$ in Utrecht and 3.4 L min$^{-1}$ LI$^{-1}$ in Hamburg, but they are distributed differently across the road (Figure
1). In Utrecht, emitting locations on level 2 roads contributed the most (50 % of emissions) to the total emissions while in
Hamburg the majority of the emissions occurred on residential roads (56 % of total emissions). This shows that the major leak
indications may happen on different road classes in different cities and there is no general relation to the size of streets between
these two cities.
In Figure 4, we compare cumulative $CH_4$ emissions for Utrecht and Hamburg to numerous US cities (Weller et al., 2019).
After ranking the LIs from largest to smallest, it becomes evident that the largest 5 % of the LIs account for about 60 % of
emissions in Utrecht, and 50 % of the emissions in Hamburg.
As mentioned above, the observed total emission rates observed on roads in urban environment in the two cities are
relatively similar when normalized by the total amount of km covered, 0.64 L min$^{-1}$ km$^{-1}$ for Utrecht and 0.4 L min$^{-1}$ km$^{-1}$ for
Hamburg (Table 2). Using these two emission factors, the observed emission rates (≈110 t yr$^{-1}$ in Utrecht and ≈180 t yr$^{-1}$ in
Hamburg) were up-scaled to the entire road network in the two cities, ≈ 650 km in Utrecht and ≈ 3,000 km in Hamburg. This
includes the implicit assumption that the pipeline network is similar to the street network. Total up-scaled emission rates based
on mobile measurements on roads in urban environment before considering attribution analysis over LI locations are 150 t yr$^{-1}$
and 440 t yr$^{-1}$ across the study areas of Utrecht and Hamburg respectively. Distributing the calculated emission rates over the
population in the city areas yields emission rates of 0.54 ± 0.15 kg yr$^{-1}$ capita$^{-1}$ for Utrecht and 0.31 ± 0.04 kg yr$^{-1}$ capita$^{-1}$ for
Hamburg (see SI, Sect. S.3.2, Figure S14).

**3.2 Attribution of CH$_4$ emissions across Utrecht and Hamburg**

Figure 5 shows the results of the isotope analysis for the 21 locations in Hamburg where acceptable Keeling plots
were obtained (see SI, Sect. S.3.3, Table S9 and Table S10). The results cluster mostly in three groups, which are characterized
by the expected isotope signatures for fossil, microbial, and pyrogenic samples as described in Röckmann et al., (2016).
Average isotope signatures for the LIs in the city of Hamburg were $\delta^{13}C$ = -52.3 ± 5.1 ‰ and $\delta D$ = -298.4 ± 30.3 ‰
for the samples characterized as microbial and $\delta^{13}C$ = -41.9 ± 1.0 ‰ and $\delta D$ = -196.1 ± 10.6 ‰ for the samples characterized
as fossil (Figure 5). One sample from the Hamburg city area displays a very high source signature of $\delta^{13}C$ = -23 ‰ and $\delta D$ =
-153 ‰. The origin of CH$_4$ with such an unusual isotopic signature could not be identified and it is considered an outlier. In
Hamburg, 10 % of the LI locations (38 % of emissions) on the north side of Elbe were sampled for isotope analysis. The lab
isotopic attributions show that the LIs with the higher emission rates are mostly caused by emission of fossil CH$_4$. 79 % of the
inferred emissions at 38 % of the LIs were identified as of fossil origin, 20 % of emissions at 54 % of the LIs as of microbial
origin (for an identified source see SI, Sect. S.3.3, Figure S15), 1 % of emissions at 8 % of LIs as of pyrogenic origin.
In Hamburg, during three passes through the new Elbe tunnel (see SI, Sect. S.3.4, Figure S16) a CH$_4$:CO$_2$ of 0.2 ±
0.1 ppb:ppm was derived for combustion-related emission. During the surveys of open roads, clear CH$_4$:CO$_2$ correlations were
observed for several LIs and an example of a measurement of car exhaust is shown in Figure S12a (see SI, Sect. S.2.6) with
CH$_4$:CO$_2$ = 1.6 ppb:ppm. Previous studies have shown relatively low CH$_4$:CO$_2$ ratios of 4.6*10$^{-2}$ ppb:ppm (Popa et al., 2014),
0.41 ppb ppm$^{-1}$ (E. K. Nam et al., 2004), and 0.3 ppb:ppm (Naus et al., 2018) when cars work under normal conditions. During
cold engine (Naus et al., 2018) or incomplete combustion conditions, the fuel to air ratio is too high, which results in enhanced
emission of black carbon particles and reduced carbon compounds, so higher CH$_4$:CO$_2$ ratios. Hu et al. (2018) reported 2 ± 2.1
ppb:ppm in a tunnel, but 12 ± 5.3 ppb:ppm [1] on roads. In addition to car exhaust, there are other combustion sources which
can affect CH$_4$ and CO$_2$ mole fractions at the street level including natural gas water heater (CH$_4$:CO$_2$ ratio of ≈ 2 ppb:ppm;
Lebel et al., 2020), restaurant kitchens, etc. Based on the CH$_4$:CO$_2$ ratio (ppb:ppm) criterion defined above (see Sect. 2.3.1),
17 % of LIs (10 % of emissions) can be attributed to combustion (see SI, Sect. S.3.4, Figure S17) with a mean CH$_4$:CO$_2$ ratio
of 3.2 ± 3.9 ppb:ppm (max = 18.7 and min = 0.8 ppb:ppm). The C$_2$:C$_1$ ratio for these LIs attributed to combustion in Hamburg
was 7.8 ± 3.5 %. In Utrecht 7 % of LIs (2 % of emissions) are attributed to combustion with a mean CH$_4$:CO$_2$ ratio of 9.8 ±
5.8 ppb:ppm (max = 16.7 and min = 3.0 ppb:ppm).
Based on the C$_2$H$_6$ signals, 64 % of the emissions (33 % of LIs) were characterized as fossil, while 25 % of emissions
(20 % of LIs) were identified as microbial. Due to low CH$_4$ and C$_2$H$_6$ enhancements, 47 % of the locations (11 % of emission)
were considered unclassified. The C2:C1 ratio for the LIs attributed to emissions from NGDNs in Hamburg study area (North
Elbe) is 4.1 ± 2.0 %. The oil production site in south-east Hamburg had a higher C$_2$:C$_1$ ratio of 7.1 ± 1.5 %.
In Utrecht, C$_2$H$_6$ was measured only during four surveys in February, April, and June 2019 (revisits of 2-day surveys
across the city center and 2 days to LIs with high emission rates) as the CH$_4$ - C$_2$H$_6$ analyzer was not available during the first
campaign. The $C_2$:$C_1$ ratios from this limited survey indicates that 93 % of emissions (69 % of the LIs across the city centre,
including combustions) are likely from fossil sources (Table 2) and 73 % of emissions (43 % of the LIs, including combustion)
out of all LIs. In Utrecht, the $C_2$:$C_1$ ratio for the LIs attributed to NGDNs is $3.9 \pm 0.8$ %.

**3.3 Quantification of $CH_4$ plume from larger facilities**

Table 3 shows the emission rate estimates from the larger facilities in Utrecht and Hamburg. $CH_4$ plumes from the
WWTP (Figure 6 and in SI, Sect. S.1.6., Table S5) were intercepted numerous times during the city transects, and the error
estimate in Table 3 represents one standard deviation of 5 sets of measurements where each measurement comprises 2-4
transects during three measurement days (12-Feb.-2018, 24-Apr.2018, and 07-Jan.-2019). Figure 7 shows an example of a fit
of a Gaussian plume to the measurements from the Utrecht WWTP. The derived distance to the source was $215 \pm 90$ m, the
hourly average wind speed was $3.5 \pm 1.1$ m s$^{-1}$ and the wind direction was $178 \pm 5$ degrees (see SI, Sect. S.1.6, Table S5).
The total emission rate of the WWTP in Utrecht was estimated at $160 \pm 90$ t yr$^{-1}$. The reported errors include stability
classes, wind speed and directions, and effective point source coordinates. Not all transects provided datasets that allowed an
adequate Gaussian fit, these were not included in total estimates from the facilities, e.g. measurements during the visits of the
harbor area in Hamburg were excluded. In Hamburg, plumes from several facilities were also intercepted several times (see
SI, Sect. S.1.6, Table S6). For a Compost and Soil Company in Hamburg we estimate an emission rate of $70 \pm 50$ t yr$^{-1}$. The
mobile quantifications at the upstream sites in Hamburg from a separator, a tank, and an oil well yield annual $CH_4$ emission
of $4.5 \pm 3.7$ t yr$^{-1}$, $5.2 \pm 3.0$ t yr$^{-1}$, and $4.8 \pm 4.0$ t yr$^{-1}$ respectively.

**4 Discussion**

**4.1 Detection and quantification**

As mentioned above (see Sect. 2.2.2), we used methods similar to the ones introduced by von Fischer et al. (2017)
and updated in Weller et al. (2019)  that were used to characterize $CH_4$ emission from local gas distribution systems in the US.
An important difference is that we did not visit each street twice in the untargeted survey, and the revisits were specifically
targeted at locations where we had found a LI during the first visit. A consequence of the different sampling strategy is that
we do not base our city-level extrapolated emissions estimates on "confirmed" LIs, as done in Weller et al. (2019) but on all
the LIs observed. In our study, 60 % of $CH_4$ LIs in Utrecht and 46 % of LIs in Hamburg were confirmed. This number may be
biased high, since we preferentially revisited locations that had shown higher LIs, and the percentage of confirmed LIs may
have been lower if we had visited locations with smaller LIs. Von Fischer et al. (2017) reported that LIs in the high emission
rate category have a 74 % chance of detection, which decreased to 63 % for the middle category and 35 % frequency for the
small category. In our study, all LIs within the high emission rate category (n = 1 and n = 2 LIs in Utrecht and Hamburg
respectively) were confirmed in both cities. Overall, the confirmation rates found in Hamburg and Utrecht were similar to the
ones reported in the US cities by von Fischer et al. (2017), suggesting that the results from both driving strategies can be
compared when we take into account an overall confirmation percentage of roughly 50 %.
In 13 US cities the "LI density" ranged from 1 LI per 1.6 km driven to 1 LI per $\approx$ 320 km driven (EDF, 2019). This
illustrates that cities within one country can be very different in their NGDN infrastructure. In Utrecht, one LI was observed
every 5.6 km of street covered and in Hamburg every 8.4 km covered. Note that we normalize the number of LIs per km of
road covered, not km of road driven, since the revisits were targeted to confirm LIs, which would bias the statistics if we
normalize by km of road driven. After accounting for the confirmation percentage of 50 %, the LI densities in Utrecht and
Hamburg become 1 LI per 11.2 km covered in Utrecht, and 1 LI per 16.8 km covered in Hamburg. When we take into account
the attributions (fraction fossil/total LIs is 43 % in Utrecht and 31 % in Hamburg), confirmed LIs from the NGDN are found
every 26 km in Utrecht and every 54 km in Hamburg. The highest 1 % of the LIs in Utrecht and Hamburg account for
approximately 30 % of emissions, emphasizing the presence of a skewed distribution of emissions. The emissions distribution
is even more skewed for these two European cities than for countrywide US cities, where approximately 25 % of emissions
comes from the highest 5 % of the LIs. Skewed emission distributions appear to be typical for emissions from the oil and gas
supply chain across different scales. For example, a synthesis study reviewing the distribution of upstream emissions from the
US natural gas system shows that in the US 5 % of the leaks are responsible for 50 % of the emissions (Brandt et al., 2016).
**4.2 Attribution**

Four different approaches were combined in Hamburg for emission source attribution, which allows an evaluation of
their molecular consistency. Figure 5 shows that measurements of the $C_2$:$C_1$, $\delta D$, and $\delta^{13}C$ provide a very consistent distinction
between fossil and microbial sources of $CH_4$. Except for one outlier with a very enriched $\delta^{13}C$ and $\delta D$ contents and no $C_2H_6$
signal, all samples that are classified as "microbial" and depleted in $\delta^{13}C$ and $\delta D$ signatures contain no measurable $C_2H_6$.
Samples that are characterized as "fossil", based on $\delta^{13}C$ and $\delta D$ signatures, bear a $C_2H_6$ concomitant signal. This strengthens
the confidence in source attribution using these tracers. The fossil $\delta^{13}C$ signature of bag samples from natural gas leaks in
Hamburg ($\delta^{13}C$ = -41.9 ± 1.0 ‰) is higher than recent reports from the city of Heidelberg, Germany ($\delta^{13}C$ = −43.3 ± 0.8 ‰
(Hoheisel et al., 2019)). This shows that within one country, $\delta^{13}C$ from NGDNs can vary from one region to another. These
numbers do not agree within combined errors, but are also not very different. $\delta^{13}C$ values of $CH_4$ from the NGDN can vary
regionally and temporally, e.g. due to differences in the mixture of natural gas from various suppliers for different regions in
Germany (DVGW, 2013). In a comprehensive study at global scale, it is also shown that how $\delta^{13}C$ values of fossil fuel $CH_4$
have significant variabilities in different regions within an individual basin (Figure 4 in Sherwood et al. (2017)).

In Hamburg both $C_2$:$C_1$ and $CH_4$:$CO_2$ analysis along with $\delta^{13}C$ and $\delta D$ signatures suggest that ≈ 50 % to ≈ 80 % of
estimated emissions (≈ 30 % and ≈ 40 % of LIs respectively) originate from NGDNs, whereas $CH_4$:$CO_2$ analysis and the
smaller sample of $C_2$:$C_1$ measurements in Utrecht suggests that the overwhelming fraction (70 - 90 % of emissions; 40 − 70 %
of LIs) originated from NGDNs. We note that although it is widely assumed that microbial $CH_4$ is not associated with ethane,
some studies have reported microbial production of ethane, so it may not be a unique identifier (Davis and Squires, 1954;
Fukuda et al., 1984; Gollakota and Jayalakshmi, 1983; Formolo, 2010). The online $C_2$:$C_1$ analysis to attribute LIs is fast and
can be used at larger scale, but with the instrument we used we were not able to clearly attribute sources with $CH_4$
enhancements of less than 500 ppb. Isotopic analysis by IRMS can attribute sources for smaller LIs (down to 100-200 ppb)
but is clearly more labor intensive, and it would be a considerable effort to take samples from all LIs observed across an urban
area. Overall, $C_2H_6$ and $CO_2$ signals are very useful in eliminating non-fossil LIs in mobile urban measurements and with
improvements in instrumentations, analyzing signals of these two species along with evaluation of $CH_4$ signals can make
process of detecting pipeline leaks from NGDN more efficient.

In Hamburg, most of the LIs were detected in the city center (Figure 1). This means that the LI density is higher than
the average value in the center, but much lower than the average value in the surrounding districts and residential areas. Many
of the LIs in the city center were attributed to combustion and microbial sources, thus they do not originate from leaks in the
NGDN. Many of the microbial LIs encountered in Hamburg are around the Binnenalster lake (see SI, Sect. S.3.3, Figure S15),
which suggests that anaerobic methanogenesis (Stephenson and Stickland, 1933; Thauer, 1998) can cause these microbial
emission in this lake, as seen in other studies focused on emissions from other lakes (e.g., DelSontro et al., 2018; Townsend-
Small et al., 2016). Microbial $CH_4$ emissions from sewage system (Guisasola et al., 2008) can also be an important source of
in this area, as seen in US urban cities (Fries et al., 2018). Fries et al. (2018) performed direct measurement of $CH_4$ and nitrous
oxide ($N_2O$) from a total of 104 sites, and analyzed $\delta^{13}C$ and $\delta D$ signatures of samples from 27 of these locations, and attributed
47 % of these locations to microbial emissions in Cincinnati, Ohio, USA.

## 4.3 Comparison to national inventory reports

In the national inventory reports, total upscaled emissions from NGDNs are based on sets of emission factors for different pipeline materials (e.g., grey cast iron, steel, or plastic) at different pressures (e.g., <= 200 mbar or >200 mbar). The reported emission factors are based on IPCC tier 3 approach (Buendia et al., 2019). However, emission estimates do not exist for individual cities including Utrecht and Hamburg. Also, it is not possible to calculate a robust city-level estimate using the nationally reported emission factors because there is no publicly available associated activity data, i.e., pipeline materials and lengths for each material, at the level of individual cities. As a result, a robust direct comparison between nationally reported emissions and our measurements, akin to a recent study in the United States (Weller et al., 2020), is currently not possible. The following juxtaposition of our estimates and national inventory downscaling to city-level is therefore provided primarily as illustration of the data gaps rather than a scientific comparison. In Utrecht, we attributed 70 – 90 % of the mobile measurement inferred emissions of $\approx$ 150 t yr$^{-1}$ to the NGDN, thus 105 – 135 t yr$^{-1}$.

The Netherlands National Institute for Public Health and the Environment (RIVM) inventory report derived an average NGDN emission factor of $\approx$ 110 kg km$^{-1}$ yr$^{-1}$ using 65 leak measurements from different pipeline materials and pressures in 2013. This weighted average ranged from a maximum of 230 kg km$^{-1}$ yr$^{-1}$ for grey cast iron pipelines to a minimum of 40 kg km$^{-1}$ yr$^{-1}$ for pipelines of other materials with overpressures <= 200 mbar (for details, see P. 130 in Peek et al. (2019)). This results in an average $CH_4$ emissions of $\approx$ 70 t yr$^{-1}$ (min = 30 t yr$^{-1}$ and max = 150 t yr$^{-1}$) for the study area of Utrecht, assuming $\approx$ 650 km of pipelines inside the ring, and further assuming that Utrecht's NGDN is representative of the national reported average (see qualifiers above). The average emissions for the Utrecht study, based on emissions factors reported for the Netherlands, is smaller by a factor of 1.5 - 2 compared to the emissions derived here. The variability factor of 5, from the reported emission (resulting from the variability in pipeline materials) highlights the need for city-level specific activity data for a robust comparison. In Hamburg, 50 – 80 % of the upscaled emissions of 440 t yr$^{-1}$ (220 – 350 t yr$^{-1}$), can be attributed to the emission from NGDN. The national inventory from the Federal Environment Agency (UBA) in Germany, reports an average $CH_4$ emission factor for NGDN from low pressure pipelines as $\approx$ 290 kg km$^{-1}$ yr$^{-1}$ (max = 445 kg km$^{-1}$ yr$^{-1}$ (grey cast iron) and min = 51 kg km$^{-1}$ yr$^{-1}$ (plastic)) based on measurements from the 1990s (Table 169 in Federal Environment Agency (2019)). Assuming $\approx$ 3000 km of pipelines in the targeted region, and further assuming that Hamburg's NGDN is representative of the national reported average (see qualifiers above), results in an estimated NGDN $CH_4$ emissions average of $\approx$ 870 t yr$^{-1}$ (min = 155 t yr$^{-1}$ and max = 1350 t yr$^{-1}$). While this study's estimate (220 – 350 t yr$^{-1}$) falls in the lower end of this range, the reported emissions variability factor of 9 (resulting from the variability in pipeline materials) highlights again the need for city-level specific activity data for a robust comparison. To put the national inventory comparison into perspective, it should be noted that GasNetz Hamburg detected and fixed leaks at 20 % of the fossil LIs in this study, which accounted for 50 % of emissions. In Utrecht and Hamburg, the natural gas consumption in our target area were retrieved through communications with LDCs. In the Utrecht and Hamburg study areas, natural gas consumption is 0.16 bcm yr$^{-1}$ (STEDIN, personal communication) and 0.75 bcm yr$^{-1}$ (GasNetz Hamburg, personal communication) respectively. The estimated emissions from NGDNs in our study is between 0.10 – 0.12 % in Utrecht and between 0.04 – 0.07 % in Hamburg of total the annual natural gas consumptions in the same area. In the US, where the majority of natural gas consumption is from residential and commercial sectors, Weller et al. (2020) reported emissions of 0.69 Tg year$^{-1}$ (0.25 - 1.23 with 95 % confidence interval), with a sum of $\approx$ 170 Tg year$^{-1}$ (U.S. EIA, 2019), showing 0.4 % (0.15 % - 0.7 %) loss from NGDNs. The US NGDNs loss is about four times larger than our reported loss in Utrecht, and is about ten times larger than the loss for Hamburg. Considering the population of Utrecht ($\approx$ 0.28 million) and Hamburg ($\approx$ 1.45 million), the natural gas consumption densities in these study areas are $\approx$ 570 m$^3$ capita$^{-1}$ yr$^{-1}$ and $\approx$ 520 m$^3$ capita$^{-1}$ yr$^{-1}$, where in the US (population $\approx$ 330 million (US Census Bureau, 2020)) the density is about $\approx$ 730 m$^3$ capita$^{-1}$ yr$^{-1}$ (see SI, Sect. S.3.2, Figure S14). This shows that annual natural gas consumption per capita in the US is about 30 % and 40 % higher than in Utrecht and Hamburg respectively. The emission per

km of pipeline in Utrecht is between 0.45 – 0.5 L min$^{-1}$ km$^{-1}$ and in Hamburg is between 0.2 – 0.32 L min$^{-1}$ km$^{-1}$. In the US,
based on 2,086,000 km km of local NGDN pipeline (Weller et al., 2020), this emission factor will be between 0.32 – 1.57 L
min$^{-1}$ km$^{-1}$. This shows higher emissions per km pipeline in the countrywide studies of US compared to just two European
cities of Utrecht and Hamburg (see qualifiers above). This can be partly explained by pipeline material, maintenance protocols,
and higher use of natural gas consumption in the US. However, the substantial variability in emission rates across US cities,
as wells as the annual variability of gas consumption over the year, again restricts a direct comparison of two cities with a
national average measured over multiple years.

Normalized LIs emissions per capita in Utrecht ($0.54 \pm 0.15$ kg yr$^{-1}$ capita$^{-1}$) are almost double the emission factor in

Hamburg ($0.31 \pm 0.04$ kg yr$^{-1}$ capita$^{-1}$). This metric may be useful to compare cities, assuming that the emission quantification
method is equally effective for different cities. $CH_4$ emissions can vary among different cities, depending on the age,
management and material of NGDNs, and/or the management of local sewer systems. In our study, we only surveyed two
cities, and the above number may not be adequate for extrapolation to the country scale (McKain et al., 2015).

## 4.4 Interaction with utilities

After the city surveys, locations with the highest emissions (high and medium categories) were shared with STEDIN

Utrecht and all LI locations were reported to GasNetz Hamburg. The utilities repair teams were sent to check whether LIs
could be detected as leaks from NGDN and fixed. The LDCs follow leak detection procedures based on country regulations
(e.g., for GasNetz Hamburg in SI, Sect. S.4.1, Table S11). GasNetz Hamburg also co-located the coordinates of the detected
reported LIs with the NGDN and prioritized repairs based on safety regulations mentioned in Table S12 (see SI, Sect. S.4.1).
This interaction with the LDCs resulted in fixing major NGDN leaks in both cities. In Utrecht the only spot in the high emission
category was reported to STEDIN, but the pipelines on this street had been replaced, which most likely fixed the leak, as it
was not found later by the gas company nor in our later survey with the $CH_4$ - $C_2H_6$ analyzer. In Utrecht, half of the LIs in the
medium category were found and repaired.

A routine leak survey (detection and repair) had been performed by GasNetz Hamburg between 1-5 months before

the campaign, for the different regions (see SI, Sect. S.4.1., Table S11). The timing of any routine detection and repair likely
influences the absolute number of LIs measured during independent mobile measurements, and the survey by GasNetz
Hamburg thus likely has influenced the absolute number of LIs measured in our campaign. We then reported the LI
latitude/longitude coordinates to GasNetz Hamburg about 4 months after our campaign. Additionally, we provided map images
of the LIs immediately after the campaign. The comparison of the number of reported LIs (and emission rates) during our
campaign with those identified by GasNetz Hamburg post-campaign assumes that the leaks continued to emit gas until they
were detected and fixed by GasNetz Hamburg (if they were detected).

Depending on how close the gas leaks are located to a building, the LDCs prioritize the leaks into four classes from

the highest to lowest priority: A1, A2, B, and C (see SI, Sect. S.4.1, Table S12). In Hamburg, both LIs in the high category
were identified as A1 gas leaks and fixed by GasNetz Hamburg immediately. Most of the Hamburg LIs that were detected and
identified as fossil are in close proximity to the natural gas distribution pipelines (see SI, Sect. S.4.2, Table S13). Investigation
of the pipeline material shows that most of NGDN emissions are due to leaks from steel pipelines (see SI, Sect. S.4.2, Table
S14), which are more prone to leakage because of pipeline corrosion (Zhao et al., 2018). Nevertheless, only 7 of the 30 LIs
(23 %) that were positively attributed to fossil $CH_4$ were detected and fixed by the LDC. If we assume that the fraction fossil
/ total LIs determined in Hamburg ($\approx 35$ %) is representative for the entire population of LIs encountered (thus also for the
ones that were not attributable), about 50 of the 145 LIs are likely due to fossil $CH_4$. The LDC found and fixed leaks at 10 of
these locations ($\approx 20$ %). A recent revisit (January 2020) to these locations confirmed that no LIs were detected at 9 out of
these 10 locations. For the 10$^{th}$ location a smaller LI was detected in close proximity, and GasNetz Hamburg confirmed that
this was a leak from a steel pipeline. The whole pipeline system on this street dates back to the 1930s and is targeted for
replacement in the near future.
In summary, about 20 % of the LIs including the two largest LIs that were attributed to a fossil source were identified
as NGDN gas leaks (see SI, Sect. S.4.2, Figure S18), and were repaired by GasNetz Hamburg, but these accounted for about
50 % of fossil $CH_4$ emissions of Hamburg, similar to what was observed in the US studies (Weller et al., 2018). Possibly,
smaller leakages that can be detected with the high sensitivity instruments used in the mobile surveys cannot be detected with
the less sensitive equipment of LDCs. Another possible explanation for the fact that the LDC did not detect more leaks may
be that reported LI locations do not always coincide with the actual leak locations, although Weller et al. (2018) reported that
the median distance of actual leak locations to the reported ones was 19 m. Combined measurements with GasNetz Hamburg
are planned to investigate why the majority of the smaller LIs reported in mobile surveys is not detected in the regular surveys
of the LDC.
The average $C_2:C_1$ ratio for LIs with a significant $C_2H_6$ signals across Hamburg was 5.6 ± 3.9 %. For the spots where
the LDC found and fixed leaks this ratio was 3.9 ± 2.6 %. Thus, some of the locations where $CH_4$ enhancements were found
were influenced by sources with an even higher $C_2:C_1$ ratio than the gas in the NGDN. One confirmed example is the very
high ratio found in exhaust from a vehicle as shown in Figure S12 (see SI, Sect. S.2.6). The abnormal operation of this vehicle
is confirmed by the very high $CH_4:CO_2$ ratio of 5.5 ppb:ppm (SI, section S2). This is more than 20 times higher than $CH_4:CO_2$
ratios of 0.2 ± 0.1 ppb:ppm observed during passages through the Elbe tunnel, a ratio that agrees with previous studies (SI,
section S2).
Repairing gas leaks in a city has several benefits for safety (preventing explosions), sustainability (minimizing GHG
emissions) and economics. Gas that is not lost via leaks can be sold for profit, but gas leak detection and repair is expensive
and is usually associated with interruptions of the infrastructure (breaking up pavements and roads). Also, as reported above,
and in agreement with the studies in US cities, for small LIs the underlying leaks are often not found by the LDCs, possibly
because their equipment is less sensitive and aimed for finding leak rates that are potentially dangerous.
Our measurements in Hamburg demonstrate that in particular smaller LIs may originate from biogenic sources, e.g.
the sewage system, and not necessarily from leaks in the NGDN. In this respect, attribution of LIs prior to reporting to the
LDCs may be beneficial to facilitate effective repair. Figure S19 (see SI, Sect. S.5) illustrates how the individual measurement
components can be efficiently combined in a city leak survey program.
**4.5 Large facilities**
The WWTP in Utrecht emits 160 ± 90 t $yr^{-1}$, which is similar to the total detected emissions (150 t $yr^{-1}$) inside the
study area of Utrecht. The emissions reported for this facility from 2010 until 2017 are 130 ± 50 t $yr^{-1}$ (Rijksoverheid, 2019),
in good agreement with our measurements. $CH_4$ emission from a single well in Hamburg was estimated at 4.4 ± 3.5 t $yr^{-1}$,
which is in the range of median emissions of 2.3 t $yr^{-1}$ reported for gas production wells in Groningen, NL (Yacovitch et al.,
2018), and average emissions of all US oil and gas production wells 7.9 ± 1.8 t $yr^{-1}$ (Alvarez et al., 2018). In Hamburg, the
emissions from a Compost and Soil Company amount to about 10 % of the total emissions in the city target region, whereas a
wellhead, a storage tank and a waste-oil separator contribute only about 1 % each. This shows that individual facilities can
contribute significantly to the total emissions of a city. The contribution of each source is dependent on infrastructure, urban
planning and other conditions in the city (e.g. age and material of pipeline, maintenance programs, waste management, sewer
system conditions, etc.), which may change the source mix from one city to another. For example, in Utrecht the WWTP is
located within our domain of study. The wastewater treatment in Hamburg most likely causes $CH_4$ emissions elsewhere.
Therefore, facility-scale $CH_4$ emissions should be reported on a more aggregated provincial or national level. For emissions
from the NGDN, the urban scale is highly relevant, as the emission can only be mitigated at this scale.

## 5 Conclusions

Mobile measurements provide a fast and accurate technique for observing and identifying even relatively small $CH_4$ enhancements (i.e., tens of ppb) across cities and are useful for detecting potential gas leaks. During our intensive measurement campaigns, 81 LIs were observed in Utrecht (corresponding to emissions of $\approx$110 t $CH_4$ yr$^{-1}$) and 145 LIs ($\approx$180 t $CH_4$ yr$^{-1}$) in Hamburg. These estimates, based on the streets covered, were then up-scaled to the total study area, using the road network map as a proxy for the length of the pipeline network which then yielded total emissions of 150 t yr$^{-1}$ and 440 t yr$^{-1}$ across the study area of Utrecht and Hamburg respectively. The isotopic signature of $CH_4$ in air samples and continuous mobile measurement of $CO_2$ and $C_2H_6$ mole fraction show that not all the LIs observed across the two cities have fossil origin. In Utrecht, $C_2:C_1$ and $CH_4:CO_2$ analyses show that 70 -90 % of emissions were fossil. In Hamburg, $C_2:C_1$, $CH_4:CO_2$, and $\delta^{13}C$-$\delta D$ analyses suggests that 50 - 80 % of emissions originate from natural gas pipelines. For the locations where samples for isotope analysis were collected, 80 % of emissions were identified as fossil. A large fraction of emissions in both cities originated from few high emitting locations. The LDC in Hamburg (GasNetz Hamburg) detected and fixed leaks at 20 % of the locations that likely due to fossil sources, but these accounted for 50 % of emissions. Large LIs were generally confirmed as gas leaks from steel pipelines. The C2:C1 ratio at the locations where gas leaks were fixed by GasNetz Hamburg was 3.9 ± 2.6 %. The mobile measurement technique is less labor and time intensive than conventional methods and can provide extensive coverage across a city in a short period. Based on our experience for the Netherlands and Germany a protocol could be developed that aids LDCs in guiding their leak detection and repair teams. The use of emission categories and source attribution can help target repair activities to the locations of large fossil emissions. Emission quantification from large facilities shows that these emissions may be equivalent to total $CH_4$ emissions from NGDN leaks in urban environments. In order to analyze discrepancies between spatial explicit measurement-based estimates as presented here with reported annual average national emissions by sectors a coordinated effort with national agencies is necessary to address the lack of publicly available activity data (e.g., pipe material) disaggregated from the national-level (e.g., at the city-level).

**Code availability:** A MATLAB® code to analyze urban surveys is available on GitHub from Maazallahi et al. (2020a).

**Data availability:** The data including in-situ measurements, GPS data, and boundary of study areas are available on the Integrated Carbon Observation System (ICOS) portal from Maazallahi et al. (2020b).

**Video supplement:** A virtual tour of the measurements is available on the Leibniz Information Centre for Science and Technology and University Library (TIB) portal from Maazallahi et al. (2020c).

**Author contributions**

H. M. performed the mobile measurements, wrote the MATLAB® code, analyzed the data, and together with T. R. drafted the manuscript. J. M. F. and M. M. contributed with air sampling and isotope analysis. D. Z. -A. and S. S. contributed to the scientific interpretation and comparison between European and US cities. Z. D. W. and J. C. v. F. facilitated comparison to US cities and contributed to the statistical analysis. H. D. v. d. G. and T. R. provided instruments, equipment, and supervised the measurements and data analysis. T. R. developed the research idea and coordinated the city campaigns. All authors contributed to the interpretation of the results and the improvement of the manuscript.

**Competing interests**: The authors declare that they have no conflict of interest.

## Acknowledgements

This work was supported by the Climate and Clean Air Coalition (CCAC) Oil and Gas Methane Science Studies (MMS) hosted by the United Nations Environment Programme. Funding was provided by the Environmental Defense Fund, Oil and Gas Climate Initiative, European Commission, and CCAC. This project received further support from the H2020 Marie Skłodowska-Curie project Methane goes Mobile – Measurements and Modelling (MEMO$^2$; https://h2020-memo2.eu/), grant number 722479. Dr. Daniel Zavala-Araiza and Dr. Stefan Schwietzke were funded by the Robertson Foundation. We thank Dr. Rebecca Fisher who supervised RHUL contribution to the isotopic analysis of Hamburg campaign. Special thanks to Prof. Stefan Bühler from the Meteorological Institute of Hamburg University and Dr. Stefan Kinne from the Max Planck Institute for Meteorology for hosting our team during the Hamburg city measurement surveys. We would like to extend our appreciation to the anonymous referees for the insightful comments which led to improvements of the manuscript. We appreciate continuous efforts from executive and management boards of GasNetz Hamburg, Dr. Luise Westphal, Michael Dammann, Dr. Ralf Luy, and Christian Feickert who facilitated productive communications, provided information on the gas infrastructure in Hamburg and organized leaks repairs with their teams in study area of Hamburg. We also thank asset manager of STEDIN Utrecht, Ricardo Verhoeve who provided information and planned leaks repairs by STEDIN in Utrecht. We thank Charlotte Große from DBI Gas and Environmental Technologies GmbH Leipzig (DBI GUT Leipzig) who helped with clarifying information on reported emission factors provided in national inventory reports. We thank the former MSc students of Utrecht University, Laurens Stoop and Tim van den Akker who helped with the measurements in Utrecht study area.

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

**Table 1: Natural gas distribution network CH$_4$ emission categories**

| Class | CH$_4$ Enhancement (ppm) | Equivalent Emission Rate (L min$^{-1}$) | Equivalent Emission Rate ($\approx$ kg hr$^{-1}$) | LI Location Colour (Figure 1, Figure 2, and Figure S14) |
|---|---|---|---|---|
| High | >7.6 | >40 | >1.7 | Red |
| Medium | 1.6-7.59 | 6 - 40 | 0.3 – 1.7 | Orange |
| Low | 0.2-1.59 | 0.5 - 6 | 0.0 – 0.3 | Yellow |


**Table 2: Measurements and results summaries across the study area, inside the ring in Utrecht and north Elbe in Hamburg**

| Study Area | | | | Utrecht (inside the Ring) | Hamburg (North Elbe) |
|---|---|---|---|---|---|
| ≈ km street driven | | Total km driven | | 1,000 km | 1,800 km |
| | | Driven once | | 220 km | 900 km |
| | | Driven more than once | | 780 km | 900 km |
| ≈ km street covered | | Total km covered | | 450 km | 1,200 km |
| | | covered once | | 230 km | 900 km |
| | | covered more than once | | 220 km | 300 km |
| LIs and emissions | | Total number | | 81 LIs | 145 LIs |
| | | LI density | | 5.6 km covered LI$^{-1}$ | 8.4 km covered LI$^{-1}$ |
| | | Total emission rate | | 290 L min$^{-1}$ | 490 L min$^{-1}$ |
| | | Average emission rate per LI | | 3.6 L min$^{-1}$ LI$^{-1}$ | 3.4 L min$^{-1}$ LI$^{-1}$ |
| | | Total emission rate per year | | 107 t yr$^{-1}$ | 180 t yr$^{-1}$ |
| LIs visited | Once | Number | | 16 LIs | 45 LIs |
| | | Emissions | | 26 L min$^{-1}$ | 68 L min$^{-1}$ |
| | | Average emission rate per LI | | 1.6 L min$^{-1}$ LI$^{-1}$ | 1.5 L min$^{-1}$ LI$^{-1}$ |
| | More than once | Number | | 65 LIs | 100 LIs |
| | | Emissions | | 264 L min$^{-1}$ | 423 L min$^{-1}$ |
| | | Average emission rate per LI | | 4.1 L min$^{-1}$ LI$^{-1}$ | 4.2 L min$^{-1}$ LI$^{-1}$ |
| Total LIs categorized based on von Fischer et al. (2017) categories | High (>40 L min$^{-1}$) | Number | | 1 LI | 2 LIs |
| | | Emissions | | 102 L min$^{-1}$ | 145 L min$^{-1}$ |
| | | Average emission rate per LI | | 101.5 (L min$^{-1}$ LI$^{-1}$) | 72.4 L min$^{-1}$ LI$^{-1}$ |
| | | % of emissions | | 35 % of total emissions | 30 % of total emissions |
| | Medium (6-40 L min$^{-1}$) | Number | | 6 LIs | 16 LIs |
| | | Emissions | | 84 L min$^{-1}$ | 176 L min$^{-1}$ |
| | | Average emission rate per LI | | 14.0 L min$^{-1}$ LI$^{-1}$ | 11 L min$^{-1}$ LI$^{-1}$ |
| | | % of emissions | | 30 % of total emissions | 36 % of total emissions |
| | Low (0.5-6 L min$^{-1}$) | Number | | 74 LIs | 127 LIs |
| | | Emissions | | 105 L min$^{-1}$ | 169 L min$^{-1}$ |
| | | Average emission rate per LI | | 1.4 L min$^{-1}$ LI$^{-1}$ | 1.3 L min$^{-1}$ LI$^{-1}$ |
| | | % of emissions | | 36 % of total emissions | 35 % of total emissions |
| Total LIs categorized based on OSM road classes | Level 1 | Number | | 6 LIs | 29 LIs |
| | | Emissions | | 5 L min$^{-1}$ | 68 L min$^{-1}$ |
| | | Average emission rate per LI | | 0.76 L min$^{-1}$ LI$^{-1}$ | 2.3 L min$^{-1}$ LI$^{-1}$ |
| | Level 2 | Number | | 16 LIs | 34 LIs |
| | | Emissions | | 145 L min$^{-1}$ | 99 L min$^{-1}$ |
| | | Average emission rate per LI | | 9.0 L min$^{-1}$ LI$^{-1}$ | 2.9 L min$^{-1}$ LI$^{-1}$ |
| | Level 3 | Number | | 3 LIs | 23 LIs |
| | | Emissions | | 10 L min$^{-1}$ | 43 L min$^{-1}$ |
| | | Average emission rate per LI | | 3.4 L min$^{-1}$ LI$^{-1}$ | 1.9 L min$^{-1}$ LI$^{-1}$ |
| | Residential | Number | | 45 LIs | 52 LIs |
| | | Emissions | | 93 L min$^{-1}$ | 274 L min$^{-1}$ |
| | | Average emission rate per LI | | 2.1 L min$^{-1}$ LI$^{-1}$ | 5.3 L min$^{-1}$ LI$^{-1}$ |
| | Unclassified | Number | | 11 LIs | 7 LIs |
| | | Emissions | | 38 L min$^{-1}$ | 6 L min$^{-1}$ |
| | | Average emission rate per LI | | 3.4 L min$^{-1}$ LI$^{-1}$ | 0.8 L min$^{-1}$ LI$^{-1}$ |
| Attribution | C$_2$:C$_1$ ratio analysis | Fossil (Inc. combustion) | % of emissions | 93 % of total emissions | 64 % of total emissions |
| | | | % of LIs | 69 % of LIs | 33 % of LIs |
| | | Microbial | % of emissions | 6 % of total emissions | 25 % of total emissions |
| | | | % of LIs | 10 % of LIs | 20 % of LIs |
| | | Unclassified | % of emissions | 1 % of total emissions | 11 % of total emissions |
| | | | % of LIs | 21 % of LIs | 47 % of LIs |
| | δ$^{13}$C and δD analysis | Fossil | % of emissions | -------------------- | 79 % of total emissions |
| | | | % of LIs | -------------------- | 38 % of LIs |
| | | Microbial | % of emissions | -------------------- | 20 % of total emissions |
| | | | % of LIs | -------------------- | 54 % of LIs |
| | | Other | % of emissions | -------------------- | 1 % of total emissions |
| | | | % of LIs | -------------------- | 8 % of LIs (Pyrogenic) |

| | | | | |
|---|---|---|---|---|
| CH₄:CO₂ ratio analysis | Combustion | % of emissions | 2 % | 10 % |
| | | % of LIs | 7 % | 17 % |
| | Other | % of emissions | 98 % | 90 % |
| | | % of LIs | 93 % | 83 % |
| $C_2$:$C_1$ ratio, $CH_4$:$CO_2$ ratio, and $\delta^{13}C$ - $\delta D$ analyses | Fossil | % of emissions | 73 % | 48 % |
| | | % of LIs | 43 % | 31 % |
| | Combustion | % of emissions | 2 % | 10 % |
| | | % of LIs | 7 % | 17 % |
| | Microbial | % of emissions | 8 % | 35 % |
| | | % of LIs | 4 % | 33 % |
| | Unclassified | % of emissions | 16 % | 7 % |
| | | % of LIs | 46 % | 19% |
| Average emission rate per km driven | | | 0.29 L min⁻¹ km⁻¹ | 0.27 L min⁻¹ km⁻¹ |
| km driven / total LIs | | | 12.5 km LI⁻¹ | 12.36 km LI⁻¹ |
| Emission factors to scale-up emissions per km covered | | | 0.64 L min⁻¹ km⁻¹ | 0.40 L min⁻¹ km⁻¹ |
| km covered per LIs | km covered / total LIs | | 5.6 km LI⁻¹ | 8.4 km LI⁻¹ |
| | km covered / red LIs | | 454.8 km LI⁻¹ | 611.4 km LI⁻¹ |
| | km covered / orange LIs | | 75.8 km LI⁻¹ | 76.4 km LI⁻¹ |
| | km covered / yellow LIs | | 6.1 km LI⁻¹ | 9.6 km LI⁻¹ |
| km road from OSM (≈ km pipeline) | | | ≈ 650 km | ≈ 3000 km |
| Up-scaled methane emissions to total roads | | | 420 L min⁻¹ (≈150 t yr⁻¹) | 1,200 L min⁻¹ (≈440 t yr⁻¹) |
| Bootstrap emission rate estimate and error | | | 420 ± 120 L min⁻¹ | 1,200 ± 170 L min⁻¹ |
| Population in study area | | | ≈ 0.28 million | ≈ 1.45 million |
| Average LIs emissions per capita (kg yr⁻¹ capita⁻¹) | | | 0.54 ± 0.15 | 0.31 ± 0.04 |
| Yearly natural gas consumption | | | ≈ 0.16 bcm yr⁻¹ | ≈ 0.75 bcm yr⁻¹ |
| Fossil emission factors | $C_2$:$C_1$ ratio attribution analysis | Average emission rate per km gas pipeline | 0.60 ± 0.2 L min⁻¹ km⁻¹ | 0.26 ± 0.04 L min⁻¹ km⁻¹ |
| | | Average emission rates per capita | 0.50 ± 0.14 kg yr⁻¹ capita⁻¹ | 0.20 ± 0.03 kg yr⁻¹ capita⁻¹ |
| | $\delta^{13}C$ and $\delta D$ attribution analysis | Average emission rates per km gas pipeline | --------------------- | 0.32 ± 0.05 L min⁻¹ km⁻¹ |
| | | Average emission rates per capita | --------------------- | 0.25 ± 0.04 kg yr⁻¹ capita⁻¹ |
| | $C_2$:$C_1$ ratio, $CH_4$:$CO_2$ ratio, and $\delta^{13}C$ - $\delta D$ analyses | Average emission rates per km gas pipeline | 0.47 ± 0.14 L min⁻¹ km⁻¹ | 0.19 ± 0.03 L min⁻¹ km⁻¹ |
| | | Average emission rates per capita | 0.39 ± 0.11 kg yr⁻¹ capita⁻¹ | 0.15 ± 0.02 kg yr⁻¹ capita⁻¹ |
| | | Average emission rates / yearly consumption | 0.10 – 0.12 % | 0.04 – 0.07 % |


**Table 3: CH₄ Emissions from larger facilities in Utrecht and Hamburg estimated with the Gaussian Plume model**

| Facility | Emission rate (t yr$^{-1}$) |
|---|---|
| Utrecht | |
| Waste Water Treatment Plant (52.109791° N, 5.107605° E) | 160 ± 90 |
| Hamburg | |
| F: Compost and Soil Company (53.680233° N, 10.053751° E) | 70 ± 50 |
| Upstream<br>D1: 53.468774° N,10.184481° E (separator)<br>D2: 53.468443° N,10.187408° E (storage tanks)<br>D3: 53.466694° N,10.180647° E (oil well) | D1: 4.5 ± 3.7<br>D2: 5.2 ± 3.0<br>D3: 4.8 ± 4.0 |


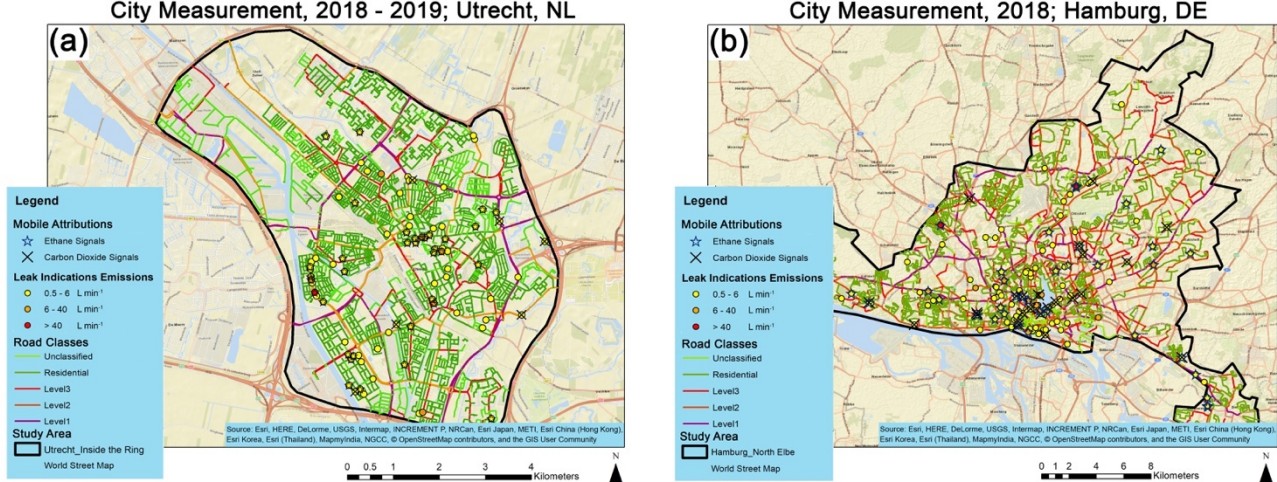

**Figure 1: Locations of significant LIs for the categories on different street classes in (a) Utrecht and (b) Hamburg. Road colors indicate the street classes according to the OSM. Black polygons show urban study areas.**


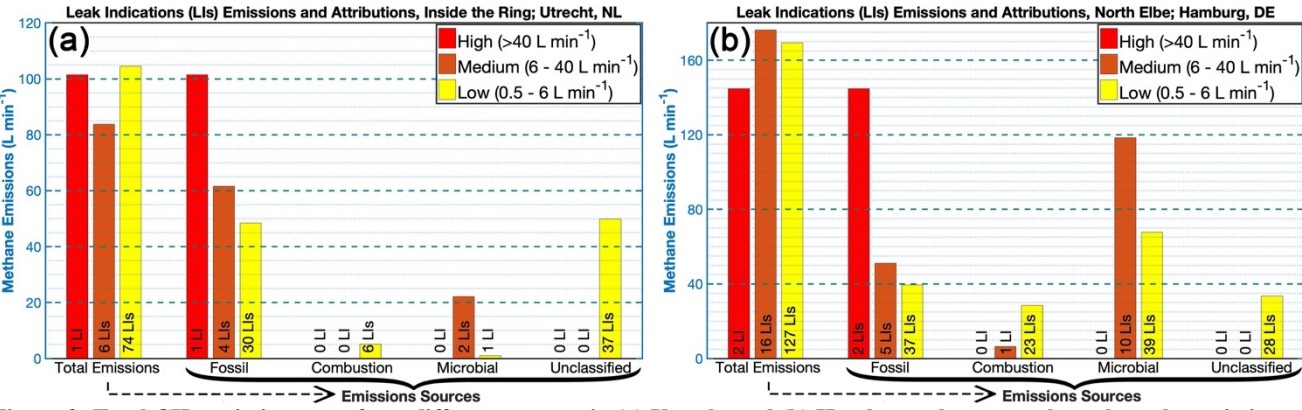

**Figure 2: Total CH₄ emission rates from different sources in (a) Utrecht and (b) Hamburg; the arrow shows how the emissions are attributed to different sources**

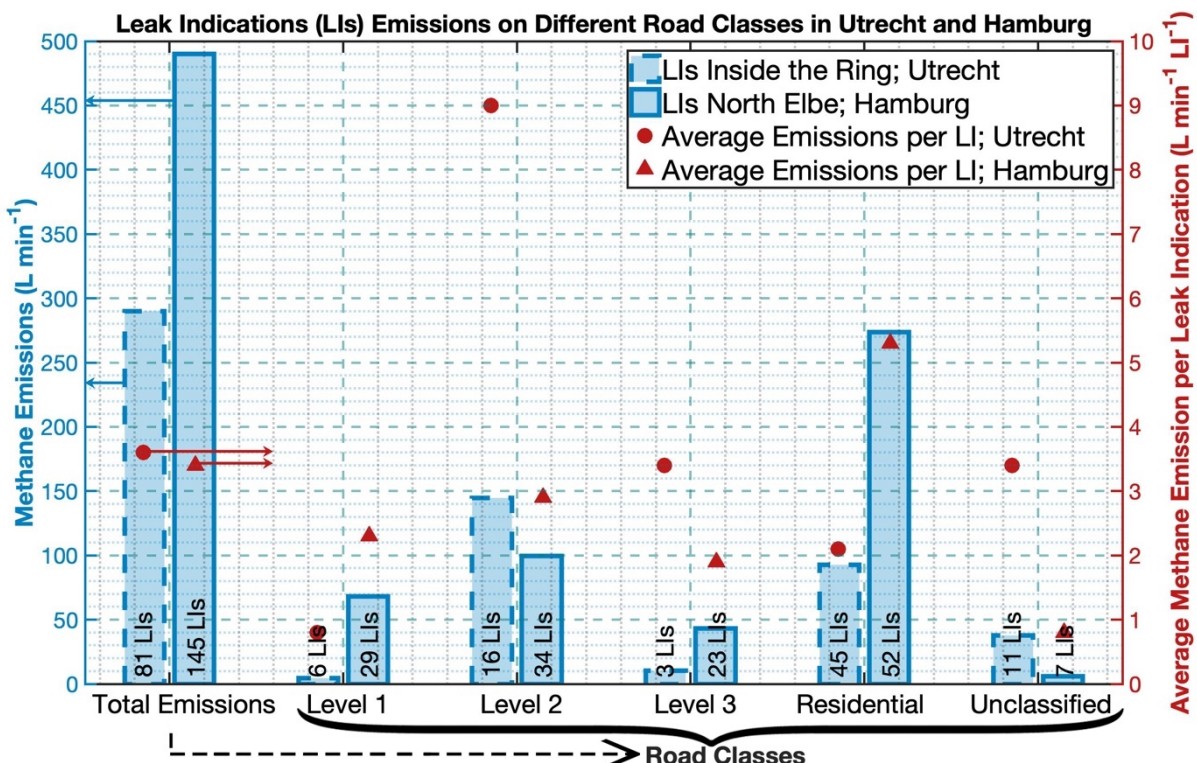

**Figure 3: Total CH₄ emissions in Utrecht and Hamburg; the arrow shows how the total emissions are distributed on different road**
**classes**

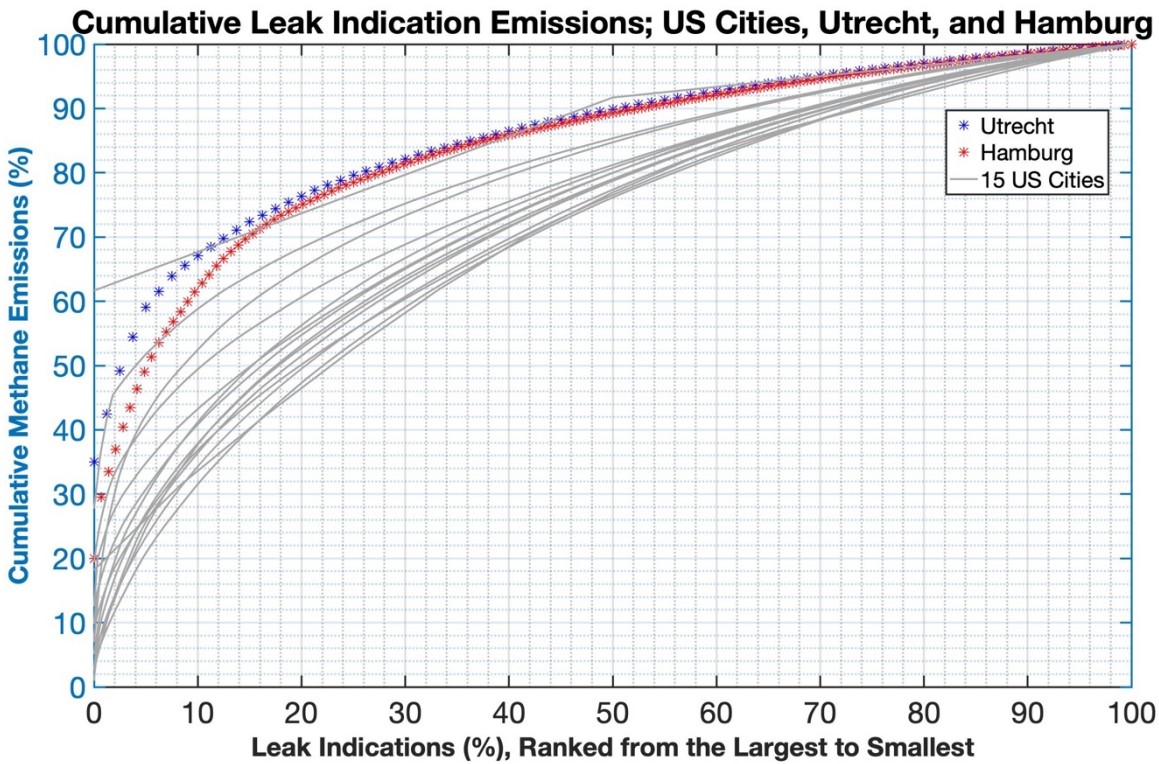

**Figure 4: Cumulative plot of CH$_4$ emissions across US cities, Utrecht, and Hamburg; datasets for the US cities are from Weller et**
**al. (2019)**

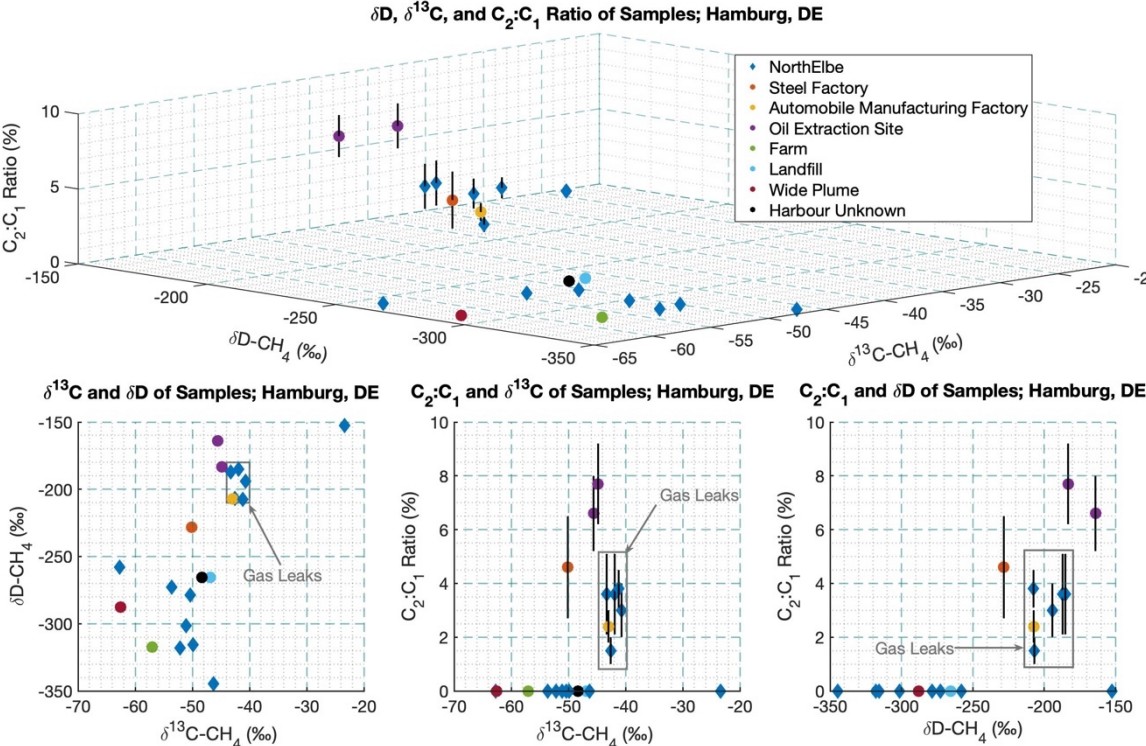

**Figure 5: Results from the attribution measurements in Hamburg: $C_2$:$C_1$ ratios, and isotopic signatures ($\delta^{13}C$ and $\delta D$) of collected**
**air samples; measurement uncertainties in $\delta^{13}C$ is 0.05 - 0.1 ‰ and in $\delta D$ is 2 - 5 ‰**

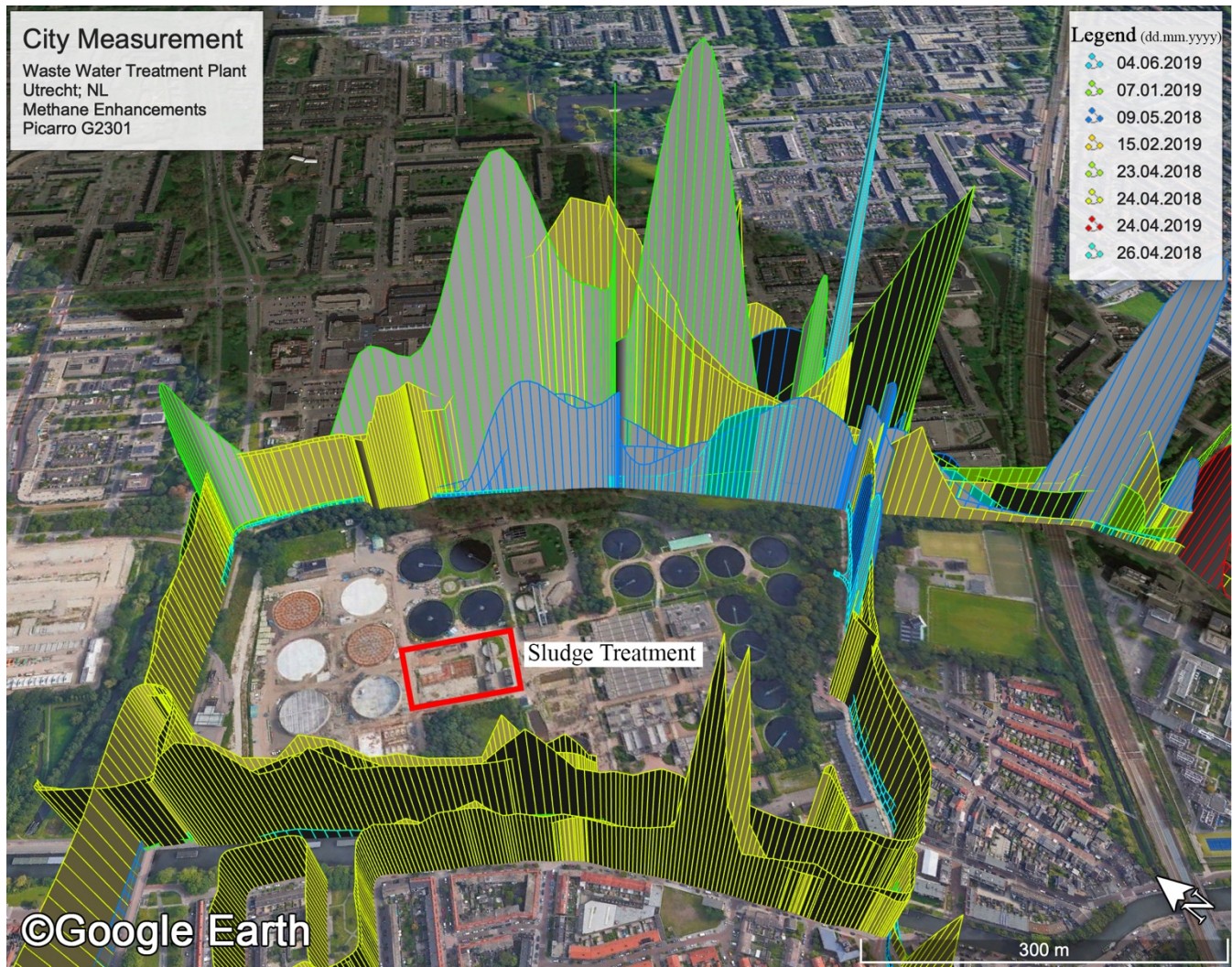

**Figure 6: CH₄ enhancements measured downwind waste water treatment plant on Brailledreef street and later used for quantifications from this facility in Utrecht; the centre of the area where the sludge treatment is located was considered as the effective CH₄ emission source, the plumes are plotted on the same scale and max CH₄ enhancement is ≈ 0.3 ppm**



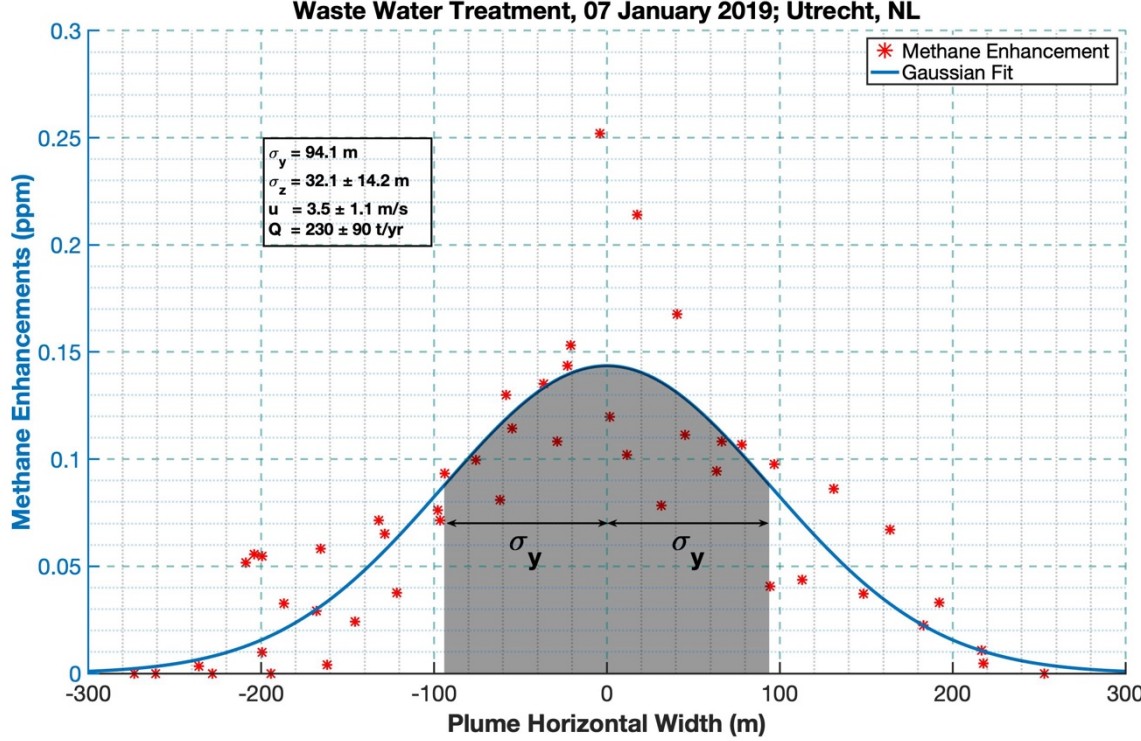

**Figure 7: Gaussian curve fitted to some transects downwind the waste water treatment plant in Utrecht**