# Peer review of "Methane mapping, emission quantification, and attribution in two"

_Atmospheric Chemistry and Physics, 2020_

## Short Comment (SC1) · 14 Sep 2020

Gasnetz Hamburg could not confirm 80 % of the LI as pipeline leaks. This issue requires further investigation. Therefore, we set up a joint project together with IMAU. The field test campaign in Hamburg is ongoing. The objective of the project is to compare leak rate estimates from mobile methods with ground measurements applying the suction method for a small sample of leaks in a real-life situation. For this reason, we request to give more explanation on that statement, e.g. by the conditional "[. . .] once the LIs were shared. Further, it must be considered that the leak detection of the gas utility and University of Utrecht did not take place at the same time (several weeks in

between). It might be possible that changing weather and soil conditions prevented finding leaks on different events. Furthermore, a "fossil leak" does not necessarily originate from a pipeline. It could also come from natural gas vehicles, thus, it is only presented for a very short time. We are highly confident, that regular LDAR (Leak Detection and Repair) is capable of finding the vast majority of leaks. Accordingly, we suggest rewording the sentence for example to "Gasnetz Hamburg could not confirm 80 % of the LI as pipeline leaks. This issue requires further Investigation."
* * *

---

## Referee Comment (RC1) · Anonymous Referee #1 · 18 Sep 2020

This study presents an extensive set of vehicle-based measurements of methane and other compounds to investigate methane emissions in Utrecht, NL and Hamburg, DE. The authors used empirical equation developed von Fischer et al. (2017) and updated by Weller et al. (2019) to estimate the emissions from the natural gas distribution network using the methane enhancements they observed during their surveys. They also tested several approaches to determine the origin of these enhancements (biogenic, thermogenic, pyrogenic) such as the isotopic signature of methane, C2/C1 ratio or CH4/CO2 ratio. Finally they used a Gaussian dispersion model to estimate methane emissions from larger sources.

Overall, despite the large ranges and uncertainties presented here, the study is a valuable contribution to the literature as the state of knowledge of urban methane is not very advanced compared to other pollutants. The measurement campaigns seems carefully done and well designed, I appreciated the authors described their interactions with the local distribution companies and showed how their work helped reducing the emissions. It was also nice to find a list of all the acronyms in the supporting information as there are so many of them in the text and I was a bit lost at first. I recommend publishing it after addressing few minor points (which are also mentioned in the detailed comments):

1) Structure: I would reorganize section 2 a bit and group all the source attribution approaches together in one subsection instead of having two subsections about isotopic analysis and information about the ratios scattered throughout the rest of the subsections. It would also make it easier for the reader if there was a table in this section summarizing these approaches and the limits used to attribute the emissions.

2) I would have liked more discussions about the uncertainty on the emissions estimated with the approach developed by Weller et al. (2019). I understand that this method is the reference for mobile surveys at the moment but the fit of the calibration curve presented in figure 4 of Weller et al. (2019) makes me wonder about the uncertainties associated with these estimates. Also, the authors used this empirical equation to estimate emissions from microbial and combustion sources whereas it was originally designed to estimate emissions from NGDN. While the classification into small, medium and large LIs depending on the maximum amplitude of the enhancement remains correct, I am not very comfortable using the empirical equation to estimate the emissions of these other sources. Biogenic emissions are very different from NGDN emissions, they are way more sensitive to atmospheric conditions (especially temperaure) and are likely to vary in time unlike NGDN emissions which should be more constant. They could also potentially be located further away than the usual roadside emissions. Figure S16 examples illustrate this: microbial emissions from the

water body are likely localized further away and the sewage system seem to emit at a higher level than the road level.

3) The presentation of the GPDM approach used to quantify the emissions from larger facilities should be reworked and expanded a bit. For examples, the authors should talk about the wind data they are using as it is a critical parameter in this approach. Did they adjust the wind direction so that the maxima of the observed plume is aligned with the maxima of the modeled plume, etc... The part about the selection of the sigma y and sigma z is also not very clear. The author should also specify here which observation they are fitting with the model (the measured concentrations? One plume at a time or all the plumes measured during all the surveys?)

Details: L18-19 and L30: Should be consistent with number notation (whether letters or numbers).

L22: This should be phrased differently, the largest emission rate in Utrecht is actually coming from the wastewater treatment plant.

L64: Typo? "high precision" is written twice in a row.

L76: Typo? A comma is missing after "(Giolo et al., 2012; Helfter et al.,2016)".

L78-80: This sentence is too vague, most methods quantify emissions methane enhancements! The authors should specify which approach they used and which type of sources they used it for.

L83: Typo? Should it be "across the urban areas in these two cities" (rather than "across the urban areas is these two cities")?

L93: Why specify the time needed to flush the cell for the G2301 but not for the G4302? Could you add a sentence about how the methane enhancements measured by the two instruments compare? This discussion is actually in the SI, the authors could add a sentence to refer it.

[Figure]

L122: What do the level 2 and 3 roads correspond to?

L137: The authors should remove "at the following links: Utrecht and Hamburg", the citation "(Maazahalli et al., 2020b)" is enough.

Section 2.4: I would merge sections 2.4 and 2.6.1 into a source attribution section that details the multiple approached used in this study. I would incorporate in this section a table summarizing the different ratios/isotopic measurements and the ranges used to distinguish between fossil, combustion, microbial and unclassified sources.

L168: Did the authors only took samples for isotopic analysis in Hamburg? Why not in Utrecht?

L179: How far are the measurements tower from the studied sites? Wind parameterization are large sources of uncertainty in Gaussian plume dispersion model, especially since wind close to the surface can be very different from the wind measured at 10 meters at these towers.

L184: "It has been demonstrated that the algorithm adequately estimates the majority of emissions from a city (Weller et al., 2018)." The authors should specify that this method was specially developed to quantify methane emissions from the natural gas distribution network. In this sentence, the authors seem to imply that they could estimate the emissions from any type of sources from a city.

L 192-194: How did the authors know about the mole percent of CH4 and C2H6 in the NGDN in Hamburg and Utrecht? Is it based on measurements or did the NG suppliers give them this information?

L196: If I understand correctly, this whole part is used to explain how you differentiate car exhaust signals from NG signals. This is not really clear, the authors should introduced it up front to help the reader follow the organization. This could probably also be moved to the source attribution section.

L204-207: I don't understand why do the authors use different approaches to estimate

the CH4 and CO2 backgrounds? This should be explain.

L229: Did the authors really need to convert decimal degrees to Cartesian coordinates in order to cluster enhancements? Doesn't it introduce additional uncertainties than directly estimating the distance between enhancements using decimal degrees?

L233: Why did the authors assigned the maximum observed enhancement to the cluster rather than a weighted average just like for the location? Wouldn't that artificially increase the emissions?

L240: The "visited at least twice" criterion in von Fischer et al. (2017) and Weller et al. (2019) was implemented to identify enhancements from the natural gas distribution network that are considered to emit continuously. I would mention that you are using another source attribution method instead.

Section 2.6.3: I was surprised that the authors did not talk about wind measurements in this section given that this is one of the biggest source of uncertainty of this technique. Maybe they should move part of section 2.5 here.

L252: What do the authors mean by "These data were evaluated using a simple point source GPDM"? What are the authors evaluating?

L252: Typo? "()" should be removed.

L265-266: The authors should be consistent with the notation: zsource (which is equal to 0 in the text) and h are to the same thing.

L276-279: This part is not very clear. Do you select sigma y and sigma z separately? Could you end up with a sigma y of a given Pasquill-Gifford stability class and combine it with a sigma z from another stability class?

L288: It would be appropriate to at least in a sentence or two explain the isotopic analysis so the reader doesn't need to go back and read these papers (which analyzer, how long were the samples measured...).

L314: Typo? "Utrecht and Hamburg correspond to" rather than "Utrecht and Hamburg were correspond to"

L321: Typo? "Figure 2" looks weird.

L332: You showed previously that different types of road had very different LI rates per km depending on cities, why didn't the authors use these road-specific emission factors to upscale their emissions?

Figure 5: Typo? "of collected air samples" instead of "of air samples collected". The authors should also show the microbial and pyrogenic clusters on these figures (L342).

L352: Typo? "combustion-related" instead of "combustion, related".

L360: Not clear which criteria for CH4/CO2 ratios the authors used to classify LIs as combustion-related in the end. CH4/CO2 > 0.2 ppb/ppm?

Figure 6: This figure is relatively difficult to interpret, it is difficult to visualize the shape of the observed plumes when they superimposed like this. It would have been interesting to see how and where you triangulated the location of the source for this site. How many sources did you find for this site? In wastewater treatment plant, the main methane source usually correspond to the sludge treatment areas that can be spotted with Google Earth.

L375-377: The definition of the error estimate is very confusing, what are the 5 sets of measurements if there were only 3 days of measurements at the wastewater treatment plant?

L395: Typo? Extra space before "74%".

L426: Typo? One of the "%" should be removed.

L413-432: The author should expand the discussion about the different source attribution approaches, is it necessary to use all of them? Which approach would the authors recommend to use in the future?

L479: Shouldn't it be the "annual natural gas leakage rate per capita" rather than the "annual natural gas consumptions per capita"?

L480: Typo? "per km of pipeline" rather than "per km pipeline"?

L491: The authors already explained several times that natural gas emissions depends on the age of the pipelines and the type of material used for these pipelines. I am not sure it is useful to repeat it here, especially since it will be discussed again later (L514).

L545-549: The authors should choose one unit for the emissions and use it for all the sources, it would make easier for the reader to compare these emissions (wastewater treatment plant in t/yr, wells in kg/h. . .).

L557: Typo? "For emissions from the NGDN, the urban. . ." rather than "For emissions from the NGDN the urban. . .".

L545-557: Did the authors also looked at the ratios of these larger facilities? It could be also be an interesting information.

Supplementary information: Section 1: "Figure S2a and Figure S2b show total length. . ." rather than "In Figure S2a and Figure S2b total length. . . are shown". Same for "In Table S1 and Table S2".

Section 2.1: Typo? Should it be "CH4-only mode, which show" (rather than "CH4-only mode. which show"). It is indeed very strange that the higher inlet measures higher methane enhancements than the bumper inlet. Would it possible that this source was located above the ground ("chimney" emissions or like the sewer pictures showed below)?

Section 2.2: What does "the ratio of the sum of CH4 enhancements (in ppb) to the sum of CO2 enhancements (in ppm)" mean? Does it correspond to the area under the plume? There is no mention of Figure S7 in the text.

Section 2.4: "Errors in wind speed are estimated to be $\pm$ 10% and for wind direction $\pm$

5°" this seems low to me considering that the wind was not measured on site but at a tower located away from the site. Table 5 caption should be better isolated from Table 4, this is a bit confusing at the moment.

Section 2.6: In Figure S10a, shouldn't the authors constraint delta13C, deltaD, C2H6 and CO2 before clustering? It would avoid clustering enhancements from different types of sources. Figure S11: caption not very precise.

---

## Referee Comment (RC2) · Anonymous Referee #2 · 8 Oct 2020

**1   Overview**

The authors present an extensive study of ground-based mobile measurements of methane and several related tracers ($C_2H_6$, $CH_4/CO_2$ ratio, $\delta^2H-CH_4$ and $\delta^{13}C-CO_2$ focused on quantifying and attributing methane emissions in two European cities, namely Utrecht and Hamburg, which both rely on subterranean pipelines as the delivery system for natural gas used in the households and otherwise. Such delivery systems are known to cause leaks that contribute to the anthropogenic global warming, and it has been demonstrated previously that fixing of these gas leaks can be a very cost-effective mean of climate change mitigation. Using a combination of in situ observations (with

[Figure]

CRDS), discrete samples collected for identified leak sources as well as Gaussian plume modelling, the authors are able to identify approximately 100 leak sources in both cities over their study period, and thanks to the robust analysis of collected data, are able to differentiate them according to emission source (natural gas distribution system / microbial sources) and the respective source strength (with just several sources responsible for large parts of total emissions). A comparison of the results against previous studies conducted in US point to potential lower specific emissions for the studied cities. The authors also attempt to upscale the measurements performed over these limited campaigns in order to compare them to the publicly available aggregated data, albeit these results should be treated with care as the dataset is limited and much more robust studies are needed to achieve this goal (which the authors accurately point out).

The authors should be commended for the impressive amount of high-quality work that was put into design, execution and data analyses during those campaigns. This is no easy task, as the study encompassed simultaneously using many state-of-the-art techniques from very different scientific fields together with a very large amount of data (both measurement and supplementary) in order to achieved the stated goals. Simultaneous analysis of several tracers and isotopic composition is of particular interest, as it shows great promise in development the methods of precise small-scale emission estimations. I find this study to be a strong contribution to the discussion in the city-scale methane emissions, and the strategy developed here seems to be promising in developing both research-targeted and operational methods for leak detection and its strength estimation.

The article does suffer however, from this wealth of data and methods, and requires multiple improvements before final publication. For example, some sections of the text require more detailed information in order for the study to be considered reproducible. Also, the treatment of uncertainty in the source estimation and Gaussian plume modelling sections should be deepened. In the second case, the sensitivity of the method to the chosen meteorological parameters should be established. In methods section

some restructuring is recommended, and the Discussion could benefit from introducing a clearly defined structure in order to appropriately focus attention.

I recommend publishing the article after addressing items listed below.

**2   Major comments:**

1. The Method description could do with an overhaul. In some places, more details need to be provided (see 'detailed comments' section below) for the experiments to be considered reproducible. In others, some information should be combined (2.6.4. and 2.4.).

2. Discussion of uncertainties in the urban emissions is very limited, with authors stating that 'We used a Bootstrap method (Nelson, 2008) to estimate 247 emission uncertainties similar to Weller et al. (2018) for the US city studies by resampling from all recorded LIs randomly 30,000 times.' No further comment is given, and in the discussion section the authors quickly skim over this and analyse the statistics of LI, without providing information on how precise those classifications might be. The method described by Wheller et al. (and earlier by Fisher et al.) relies heavily on assumptions regarding the distance to the source and (calibration using control releases) it stands to reason that this simplistic approach must produce very large uncertainties if not supported by multiple measurement repetition. This is critically important here as the data from limited detections is interpreted and up-scaled. As it is now, it is not possible to get a realistic impression about the numbers given, and puts the resulting data analysis in question. The discussion of uncertainty and potential biases should therefore be expanded.

3. The authors state that 'Emissions from facilities show significant contributions to the total emissions in both cities.' (supplement), but many details on the method

used for estimating them are missing. This section needs to be expanded and more info should be given about the analysis as well as the uncertainty estimation and sensitivity of the method to the stated assumptions. For example, the use of measurement data from distant towers in order to drive the transport model raises an eyebrow, as these are critical for calculating the emission rate. What was the average distance between the tower and the measurement location and what was the elevation of that measurement? Are the wind speed and direction uncertainties reasonable? What about the elevation of the source, which is only very briefly discussed in the supplement? In the end, the reader should have a comprehensive view on whether the method is able to provide good emission estimation in a given setting, and at the moment the result with error bars (on the order of $\pm$ 50 % 1-$\sigma$) suggest it is not. This should also be discussed in more detail.

4. The discussion section would benefit from introducing subsections to provide focus for specific items under discussion.

5. I find the overall quality of language very high, yet there are multiple minor deficiencies that still need to be addressed. Below I have listed some of them. I believe that this is mostly due to heavy editing during manuscript preparation, and I ask the authors to take special care of that issue before resubmitting.

6. The supplement is large and – I'm sorry to say that – poorly edited (tables are too large - the font can safely be made much smaller; order of figures and sections does not correspond to the manuscript reference order). In some cases, it is a source of important information that is also in some places missing from the main text (already mentioned section 2.4. and corresponding S.2.4, figure S16). If the authors want to keep some technical details apart from the main text (understandable with that much material), then I would ask to consider putting the more important parts in the Annex, in order to a) maintain the high editing standard and thus make reading easier, b) keep the important information together with the text. At the very minimum the editing of the supplement needs to be improved.

**3   Detailed comments:**

Line numbers are given for identification. Comments for figures are given at the end of the list.

L25, also later in the text: ACP requires exponential notation of units, consult the 'manuscript preparation' on the ACP website for details (www.atmospheric-chemistry-and-physics.net/for_authors/manuscript_preparation.html)

L35: I'd suggest putting ppm outside of parentheses and mole fraction inside, as the ppm/ppb notation is the dominant one in the manuscript.

L46-49: Sentence needs rephrasing

L60-62: This paragraph does not fit well here, would be better if info given as part of previous or next.

L64: 'high precision' used twice

L75-77: What were the main findings from these studies? Specifically, it would be good to comment on whether these methods can be useful for up-scaling.

L78: 'We quantified emissions in this study using measured $CH_4$ enhancements above background, which were detected' - This needs revision; also, it feels like Weller et al. 2019 should already be quoted here, perhaps something like: 'In this study, we have quantified the $CH_4$ emissions using the method described by Weller et al., who demonstrated . . .'

L91: Was the reproducibility tested by the authors? Picarro currently gives 0.5 ppb for 5 s raw data. If the reproducibility was tested by the authors, please provide some details on the testing (either reference, or brief description of the experiment). Was the water correction modified or the factory settings were used? Please state that explicitly and also provide information if necessary.

L99: Discard 'about' or the approximation sign

L100-101: Similar to comment for L91, please provide more info.

L104: Info on how the delay was calculated should be given here, but can be found later in L202. Please combine both (see major comment no. 1)

L111-112: Please spell out the main findings of the discussed comparison. Also, the reference to annex section number (S.2.) where it is discussed should be present (next to table S3 ref.). In general, sections of supplement should be referenced and not only tables or figures from it.

L129: When reading the sentence for the first time I have understood that the gas pipeline network corresponds to the street network 1:1. Is that correct, or the general coverage of municipal areas is meant? Please clarify.

L137: discard 'at the following links: Utrecht and Hamburg'

L147-148: Please briefly explain how the vehicles can be methane sources (with reference for subsequent discussion further in the text).

L148: Please state clearly how many revisits were usually made.

L159-161: Have any cases where new leaks have occurred in-between surveys been observed?

L166-167: Parentheses missing? 3 L bag for a price of 2 L bag is too good to be true.

L168-169: More details on sampling are needed. Was the data collection stationary

or also mobile? How was the plume / non-plume location determined? What was the flushing time? Was the sample dried? How?

L173: See major comment no. 3.

L191-192: Case shown in S5 is special and I strongly recommend to remove it from here and discuss later. As it is now, the text does not explain it, and thus may imply that all the cars are potential sources of $CH_4$, which is certainly not the case.

L194: Consider providing these standards in the parentheses or the supplement section S.7.

L196: How many such cases were observed? Could they be important for the overall budget?

L198: Please state the reasons for this exclusion, briefly.

L199-200: Just a small comment, no action needed: I don't see the benefit of this artificial increase of the data frequency. This brings no new information at the cost of tripling of the data that needs to be processed.

L200-201: Wording. If the time was just converted to UTC, then calling it 'a correction' is not warranted. Consider changing to: 'Following the interpolation step, the data was converted to UTC, and subsequently corrected for ...'

L202: About the delay time estimation: 5–30 seconds is a very broad range. Were the ranges so variable for both instruments, or was it 5 for one and 30 for the other? Also, how was the pulse generated? Can you estimate precision of that delay estimation (even grossly)?

L204: Reference order needs correction. Previous reference supplement figure was S5 (L190).

L207: In $CO_2$ signal (Fig. S8), it can be clearly seen that the background line is sometimes higher than the observed signal. Since this plot is about the background, it

would be good to change the limits of y axis to make the calculated background visible clearer, especially for methane. Please give some comment about the possible negative enhancements after subtracting such background (can it affect the estimation of emissions?).

L217: Please add 'peak' after enhancement, to make it clear that it's not about the release height.

L237: Wording. Why should results from different cities be comparable? The authors clearly mean that the analysis software used on a given dataset should be comparable. Please clarify. Actually, this whole paragraph can be limited to information that 'Our software was compared to analysis tools developed by CSU (von Fisher et al. 2017, Weller et al. 2019) and no significant differences were observed (see SI, section S.2.7)'.

L251: Erase 'areas'.

L252: Erase empty parentheses.

L253 and L384: 'drive-by'- I propose 'mobile'. I was surprised to find it used in Fisher et al. (albeit only once), as in U.S. this word is sometimes used to describe something much more nefarious then GHG observations.

L254: 'We report (. . .)' – Unclear what is meant in this sentence, please rephrase it.

L256-257: Erase 'both' and 'each day's'

L269-274: How was the release height determined? How is the uncertainty of this determination included in the uncertainty of emission?

L283: This section should be combined with 2.4.

L286: Info on the isotopic scales used in this study needs to be given.

L293-294: Please provide explanation on why these particular ranges were selected.

For signatures, specifically also provide references supporting the choice of isotopic signatures. Please keep in mind that for fossil fuel signatures, figure 7 from Rockmann et al. 2016 doesn't give a full picture – see e.g. Sherwood et al. 2017 for a broad overview of isotopic signatures for fossil methane.

L314-315: 'were correspond' – corresponded

L315: What is the uncertainty here? This relates to major comment 2.

L378: Why uncertainty given only for wind direction?

L394: von Fisher – V should be capitalised at the start of the sentence (Von Fisher).

L404: 'About 50 %' - please give the specific number that was used in the calculation.

L420-422: These numbers are in fact very similar. The variability of $\delta^{13}C$ in the natural gas can be quite substantial. See e.g. Fig 4 in Sherwood et al, 2017.

L426: % used twice

L433: Reference to Figure 1?

L436-438: a) Fig S16 also points to the local sewage system as potentially important source, but this is not mentioned here. b) Please be more precise in the argument here - i.e. explain why measurements around the lake point to anaerobic methanogenesis specifically. Linking to a), please include info on the potential role of the sewage system if needed. Is it possible that the sewage is seeping into the lake?

L443: 'because there is no publicly available activity data for associated activity data' – please rephrase.

L452: Too many parentheses. I suggest '(...) 40 kg km$^{-1}$ yr$^{-1}$ (for other material, p < 200 mbar; see p. 130 in Peek et al., 2019, for details)

L473: 'credibility interval' – confidence interval

L491: '(...) factors can be gas pipeline age and material, sewer system.' Part of sentence missing? Please rephrase.

L531: 'were' – where

L533: 'as shown' used twice.

L542-544: The scheme from S18 cannot be treated as a 'protocol' without a proper description of the method. In reality, it describes the main components of the method applied in the study, so in fact the manuscript itself is more of such a protocol. As it stands now, consider either expanding the description in the supplement (so gas companies might actually use it as a protocol) or discard it altogether.

L561: 'corresponding to emissions of about 107 t $CH_4$ / yr' – exactly 110 t yr$^{-1}$ is given in L332.

L562: Please state the method, e.g. 'These estimates, based only on the studied area, were then up-scaled for the total municipal area, using the road network map as a proxy to (...)'

L567: 'were from' - I suggest 'originated from'

Figure 2: a) Please fix the x axis description - extra arrow unnecessary. b) the plot is cropped in the lower part, by several pixels. c) extra grid dashed lines (green) are unnecessary, make the labels difficult to read.

Figure 3: a) again, the arrow from total emissions to 'Road classes' seems unnecessary. b) please explain the arrows from the plot in the caption.

Figure 4: I recommend plotting all US cities in a single colour (grey?) and simply label the line as ('15 US Cities', Weller et al.) or similar. The colours are indistinguishable anyway.

Figure 5: Excellent plot! a) Please add comment about the uncertainty of $\delta^{13}C$ and $\delta^2H$ signatures in the caption (they were only plotted for C2/C1. I also softly suggest to label the plots with a-b-c-d and move the labels into figure caption. b) On previous plot

the units were placed in parentheses, consider keeping notation consistent.

Figure 6: a) Plumes of what? b) Please also provide information on whether the peaks are on the same scale; if yes, then what is the plotted range of mole fractions (if those are mole fractions)?

**4  References**

2017, Sherwood et al., Global Inventory of Gas Geochemistry Data from Fossil Fuel, Microbial and Burning Sources, version 2017, Earth Syst. Sci. Data, 9, 639–656, 2017 https://doi.org/10.5194/essd-9-639-2017

---

## Author Comment (AC1) · 10 Nov 2020

This document provides our answers to Luise Westphal for "**Methane mapping, emission quantification, and attribution in two European cities; Utrecht, NL and Hamburg, DE**"

We thank Luise Westphal for the comment; please find our reply below. In the following, the short comment is in normal black text, our replies are in normal blue, and changes is the manuscript are in *blue italic* format.

Gasnetz Hamburg could not confirm 80 % of the LI as pipeline leaks. This issue requires further investigation. Therefore, we set up a joint project together with IMAU. The field test campaign in Hamburg is ongoing. The objective of the project is to compare leak rate estimates from mobile methods with ground measurements applying the suction method for a small sample of leaks in a real-life situation. For this reason, we request to give more explanation on that statement, e.g. by the conditional "[…] once the LIs were shared. Further, it must be considered that the leak detection of the gas utility and University of Utrecht did not take place at the same time (several weeks in between). It might be possible that changing weather and soil conditions prevented finding leaks on different events. Furthermore, a "fossil leak" does not necessarily originate from a pipeline. It could also come from natural gas vehicles, thus, it is only presented for a very short time. We are highly confident, that regular LDAR (Leak Detection and Repair) is capable of finding the vast majority of leaks. Accordingly, we suggest rewording the sentence for example to "Gasnetz Hamburg could not confirm 80 % of the LI as pipeline leaks. This issue requires further Investigation."

We acknowledge the fact that no leaks were found at a large number of locations could have several reasons. Therefore, we changed the respective sentence to (see Abstract, L30-32): *The largest leaks were located and fixed quickly by GasNetz Hamburg once the LIs were shared, but 80 % of the (smaller) LIs attributed to the fossil category could not be detected/confirmed as pipeline leaks. This issue requires further investigation.*

We want to specify that in our algorithm emissions from vehicles are identified (when attribution is possible) by using the co-emitted species $C_2H_6$ and $CO_2$. The 80% of LIs where no leaks were detected by GNH refers to the LIs that we have attributed to the category "fossil", which should be specific for leaks from the NGDN.

---

## Author Comment (AC2) · 10 Nov 2020

This document provides our answers to the anonymous referee #1 for "**Methane mapping, emission quantification, and attribution in two European cities; Utrecht, NL and Hamburg, DE**"

We would like to extend our appreciation to the anonymous referee for the insightful and point-by-point comments from careful reading of our paper. In the following sections, comments from the referee are provided in normal black text, our replies are in normal blue, and changes is the manuscript are in *blue italic* format.

**1 Overview**

This study presents an extensive set of vehicle-based measurements of methane and other compounds to investigate methane emissions in Utrecht, NL and Hamburg, DE. The authors used empirical equation developed von Fischer et al. (2017) and updated by Weller et al. (2019) to estimate the emissions from the natural gas distribution network using the methane enhancements they observed during their surveys. They also tested several approaches to determine the origin of these enhancements (biogenic, thermogenic, pyrogenic) such as the isotopic signature of methane, $C_2/C_1$ ratio or $CH_4/CO_2$ ratio. Finally, they used a Gaussian dispersion model to estimate methane emissions from larger sources.

Overall, despite the large ranges and uncertainties presented here, the study is a valuable contribution to the literature as the state of knowledge of urban methane is not very advanced compared to other pollutants. The measurement campaigns seems carefully done and well designed, I appreciated the authors described their interactions with the local distribution companies and showed how their work helped reducing the emissions. It was also nice to find a list of all the acronyms in the supporting information as there are so many of them in the text and I was a bit lost at first. I recommend publishing it after addressing few minor points (which are also mentioned in the detailed comments):

**1 General comments**

1) Structure: I would reorganize section 2 a bit and group all the source attribution approaches together in one subsection instead of having two subsections about isotopic analysis and information about the ratios scattered throughout the rest of the subsections. It would also make it easier for the reader if there was a table in this section summarizing these approaches and the limits used to attribute the emissions.
  - We have followed this suggestion and section, Section 2 is now rearranged to the following sub-sections: 2.1. Data collection and instrumentation, 2.2. Emission quantification, and 2.3. Emission attribution.
  - In the "data collection" section we keep sub-sections of mobile measurement of $C_2H_6$ and $CO_2$ separated from sampling for isotopic analysis as the analytical techniques for these two attribution methods are very different. Nevertheless, we now combine data evaluations of $C_2H_6$, $CO_2$, and isotopic analysis in sub-section 2.3.

2) I would have liked more discussions about the uncertainty on the emissions estimated with the approach developed by Weller et al. (2019). I understand that this method is the reference for mobile surveys at the moment but the fit of the calibration curve presented in figure 4 of Weller et al. (2019) makes me wonder about the uncertainties associated with these estimates. Also, the authors used this empirical equation to estimate emissions from microbial and combustion sources whereas it was originally designed to estimate emissions from NGDN.

While the classification into small, medium and large LIs depending on the maximum amplitude of the enhancement remains correct, I am not very comfortable using the empirical equation to estimate the emissions of these other sources. Biogenic emissions are very different from NGDN emissions, they are way more sensitive to atmospheric conditions (especially temperature) and are likely to vary in time unlike NGDN emissions which should be more constant. They could also potentially be located further away than the usual road side emissions. Figure S16 examples illustrate this: microbial emissions from the water body are likely localized further away and the sewage system seem to emit at a higher level than the road level.

- Yes, the uncertainties of the quantification algorithm introduced by von Fischer et al. (2017) and improved by Weller et al. (2019) for individual LIs are large. It is indeed evident from figure 4 of Weller et al. (2019) that individual LIs can be strongly under- or overestimated. The rationale is that when a complete city is surveyed, the contribution from the different LI categories and the total emission rate can be estimated more precisely. Following the request from the referee, we have explicitly clarified this better in the revised version (see Sect. 2.2.2, L293-302).

- We acknowledge that the algorithm in the original papers was designed to quantify pipeline leaks. Attempts to exclude emissions from other sources were to restrict the spatial extent of a $CH_4$ plume to <160 m, to require a minimum enhancement of 10% above background, and to require multiple detections. In our study, we add explicit attribution for many LIs by evaluating the co-emitted (or not) tracers $C_2H_6$ and $CO_2$. Rather than simply flagging and neglecting these LIs for the quantification of potential pipeline leaks, we use the same algorithm for quantifying emissions from other categories, namely microbial processes and combustion. We thank the referee for pointing out that this needs to be spelled out specifically in the paper. Whereas biogenic emissions, e.g. from the sewage or wastewater systems, will be released through manholes, waste water drains and other cavities that are also important for the release of $CH_4$ to the atmosphere, combustion related emissions may come from vehicles or houses and thus have different release pathways. The quantification of emissions from these source categories is thus based on the assumption that the same conversion equation can be used. This is especially the case for microbial emissions from manholes, where the enhancements and distance of the emission release point is very similar to NGDN leaks, hence the emission quantification approach applies for both source categories. In the revised version, we have stated this explicitly. We now suggest that the number of detected enhancements that have been attributed to microbial and combustion sources is more reliable than the emission rates.

- We have considered at an early stage of our research to use the information of co-emitted species (especially $C_2H_6$) to focus the paper only on NGDN leaks but decided against this. We feel the biogenic emissions are part of the overall anthropogenic urban $CH_4$ emission and, while being more uncertain, it is relevant to have an approximation of the importance in relation to NGDN. We thank the reviewer for reminding us that we need to be extra careful with this category of emissions (and we agree) but still think it is valuable information.

3) The presentation of the GPDM approach used to quantify the emissions from larger facilities should be reworked and expanded a bit. For example, the authors should talk about the wind data they are using as it is a critical parameter in this approach. Did they adjust the wind direction so that the maxima of the observed plume is aligned with the maxima of the modeled plume, etc. The part about the selection of the sigma y and sigma z is also not very clear. The author should also specify here which observation they are fitting with the model

(the measured concentrations? One plume at a time or all the plumes measured during all the surveys?)

- The relevant section has been expanded to provide more information. The reason for using data from the two mentioned towers in Utrecht and Hamburg is that the online data logging setup failed to continuously record all the local wind measurements during the surveys. The distance of the towers to the facilities ranges from 8 to 20 km, and indeed these distances introduce extra uncertainties in emission quantification, mainly related to wind speed (see Sect. 2.1.5). When we compare the data that were recorded on the vehicle with the tower, we derive a difference in wind speed of ± 10 %. After considering the remarks of the reviewer, we increase this to a more conservative error estimate of ± 30 % (see Sect. 2.2.3, L352-359).
- Regarding the wind direction (e.g. the oil wells), for several sources the emission point is relatively certain and confirmed by analysis of Google Earth images. In addition, we passed several sources during different wind conditions and did a "triangulation" based on the observed plumes and wind data (see Sect. 2.2.3, L324-333).
- For explanation on how sigma_y and sigma_z were derived, see the detailed comment below.
- Regarding the last question, we fit each plume individually (since they were often observed during different days at different locations) and average the individual results. This has now been specified (see Sect. 2.2.3, L343-344).

**3 Detailed comments**

L18-19 and L30: Should be consistent with number notation (whether letters or numbers).
- Done

L22: This should be phrased differently, the largest emission rate in Utrecht is actually coming from the wastewater treatment plant.
- This sentence focuses on the emissions from LIs found in the normal mobile surveys, and doesn't include emissions from larger facilities.

L64: Typo? "high precision" is written twice in a row.
- Done

L76: Typo? A comma is missing after "(Giolo et al., 2012; Helfter et al., 2016)".
- Done

L78-80: This sentence is too vague, most methods quantify emissions methane enhancements! The authors should specify which approach they used and which type of sources they used it for.
- The respective sentence was changed as follows (see Sect. 1, L88-90): *In this study, we quantified LIs emissions using an empirical equation from Weller et al. (2019), which was designed based on controlled release experiments from von Fischer et al. (2017), to quantify ground-level emissions locations in urban area such as leaks from NGDN.*

L83: Typo? Should it be "across the urban areas in these two cities" (rather than "across the urban areas is these two cities")?
- Done

L93: Why specify the time needed to flush the cell for the G2301 but not for the G4302? Could you add a sentence about how the methane enhancements measured by the two instruments compare? This discussion is actually in the SI, the authors could add a sentence to refer it.

- This sentence describes the smoothing effect in the cell of G2301. Information on G4302 is added to the revised manuscript as follows (see Sect. 2.1.1, L110-111): *The flow rate is 2.2 L min$^{-1}$ and the volume of the cell is 35 ml (operated at 600 mb, thus 21 ml STP) so the cell is actually flushed in 0.01 s, which means that mixing is insignificant given the 1 s measurement frequency of the G4302.*

L122: What do the level 2 and 3 roads correspond to?

- Information on the level 2 and 3 are now added to the paragraph and the respective sentence was changed as follows (see Sect. 2.1.2, L151-154): *Level 1 roads are primarily larger roads connecting cities, level 2 roads are the second most important roads and part of a greater network to connect smaller towns, level 3 roads have tertiary importance level and connect smaller settlements and districts.*

L137: The authors should remove "at the following links: Utrecht and Hamburg", the citation "(Maazahalli et al., 2020b)" is enough.

- Done

Section 2.4: I would merge sections 2.4 and 2.6.1 into a source attribution section that details the multiple approached used in this study. I would incorporate in this section a table summarizing the different ratios/isotopic measurements and the ranges used to distinguish between fossil, combustion, microbial and unclassified sources.

- Section two is now rearranged to three main sections: '2.1. Data collection and instrumentation', '2.2 Emission quantification', and '2.3. Emission attribution'. In Sect. 2.1 we present information on the mobile measurement of $CO_2$ and $C_2H_6$ and sample collection. In Sect. 2.2, we provide information on emission quantification of LI and larger facilities separately. In Sect. 2.3, we combine the attribution approaches as suggested by the referee.

L168: Did the authors only took samples for isotopic analysis in Hamburg? Why not in Utrecht?

- Due to time and budget limitations, we were only able to take a sufficient number of samples for attribution in Hamburg.

L179: How far are the measurements tower from the studied sites? Wind parameterization are large sources of uncertainty in Gaussian plume dispersion model, especially since wind close to the surface can be very different from the wind measured at 10 meters at these towers.

- The distance of the towers to the facilities ranges from 8 to 20 km. The reason for using the tower data was that wind data were not logged by the 2-D anemometer continuously (instrument failure) and the limited data collected were not sufficient to analyze emissions from these facilities.

L184: "It has been demonstrated that the algorithm adequately estimates the majority of emissions from a city (Weller et al., 2018)." The authors should specify that this method was specially developed to quantify methane emissions from the natural gas distribution network. In this sentence, the authors seem to imply that they could estimate the emissions from any type of sources from a city.

- The respective sentence was changed as follows (see Sect. 2.2.2, L246-L249): *This algorithm was designed to quantify $CH_4$ emissions from ground-level emission release locations within 5-40 m from the measurement (von Fischer et al., 2017), such as pipeline leaks and has been demonstrated that the algorithm adequately estimates the majority of those emissions from a city (Weller et al., 2018).* As mentioned in the reply to the general comment 2, in this study we use the same algorithm to provide indicative estimates of emission rates of microbial and combustion emissions as well. We note that emission pathways to the atmosphere are partially different for such emissions. Therefore, the emission rates should be seen as indicative, whereas the LI numbers from the different categories are more reliable.

L192-194: How did the authors know about the mole percent of $CH_4$ and $C_2H_6$ in the NGDN in Hamburg and Utrecht? Is it based on measurements or did the NG suppliers give them this information?

- This data was indeed provided by the network operators. This is now indicated as "personal communication" in the manuscript (see Sec. 2.3.1, L3612-364) as follows: *During the Utrecht campaign, the overall mole fraction of $CH_4$ and $C_2H_6$ in the NGDN was ≈ 80 % and ≈ 3.9 % (STEDIN, personal communication) and in Hamburg the mole fraction of $CH_4$ and $C_2H_6$ in the NGDN was about ≈ 95 % and ≈ 3.4 % (GasNetz Hamburg, personal communication) respectively.*

L196: If I understand correctly, this whole part is used to explain how you differentiate car exhaust signals from NG signals. This is not really clear, the authors should introduced it up front to help the reader follow the organization. This could probably also be moved to the source attribution section.

- Done

L204-207: I don't understand why do the authors use different approaches to estimate the $CH_4$ and $CO_2$ backgrounds? This should be explained.

- The background determination method for $CH_4$ from Weller et al. (2019) was used to stay compatible with the quantification algorithm for the urban studies. But this algorithm doesn't include background extraction for $CO_2$ and here we chose background detection methods commonly used in the literature (see Sect. 2.2.1 and Figure S7 in SI, Sect. S.2.1).

L229: Did the authors really need to convert decimal degrees to Cartesian coordinates in order to cluster enhancements? Doesn't it introduce additional uncertainties than directly estimating the distance between enhancements using decimal degrees?

- The constraint for clustering based on the von Fischer et al. (2017) algorithm is 30 m, thus we need to have the data in metric system. There are many ways to convert the decimal degrees to metric system, we used this way as it gave a very good one-to-one correlation with $R^2$=1.00 when we compared output of the equation we used for converting to cartesian system to e.g. EPSG:32632 projection. Easy implementation of this equation in the code we wrote to evaluate the data is another advantage of using this method to convert decimal degrees to metric system.

L233: Why did the authors assigned the maximum observed enhancement to the cluster rather than a weighted average just like for the location? Wouldn't that artificially increase the emissions?

- This follows the algorithm from Weller et al. (2019).

L240: The "visited at least twice" criterion in von Fischer et al. (2017) and Weller et al. (2019) was implemented to identify enhancements from the natural gas distribution network that are considered to emit continuously. I would mention that you are using another source attribution method instead.

- The following sentence was added to the paper (see Sect. 2.2.2, L287-288): *Instead, we used explicit source attribution by co-emitted tracers.* This topic was also discussed earlier in (see Sect. 2.3, L189-193): *Due to time and budget restrictions, it was not possible to cover each street at least twice, as done for the US cities. After evaluation of the untargeted first surveys that covered each street at least once, targeted surveys were carried out for verification of observed LIs and for collection of air samples at locations with high $CH_4$ enhancements. The rationale behind this measurement strategy is that if an enhancement was not recorded during the first survey, it obviously cannot be verified in the second survey.*

Section 2.6.3: I was surprised that the authors did not talk about wind measurements in this section given that this is one of the biggest source of uncertainty of this technique. Maybe they should move part of section 2.5 here.

- We acknowledge that the wind speed is a large source of uncertainty in the GPDM. The section is now revised and more information has been added in Sect. 2.2.3. See our answer to general comment 3.

L252: What do the authors mean by "These data were evaluated using a simple point source GPDM"? What are the authors evaluating?

- We meant that the data collected downwind the larger facilities were analyzed using GPDM. The respective sentence was changed as follows (see Sect. 2.2.3, L306): *We applied a standard point source GPDM (Turner, 1969) to quantify methane emissions from these larger facilities.*

L252: Typo? "()" should be removed.

- Done.

L265-266: The authors should be consistent with the notation: zsource (which is equal to 0 in the text) and h are to the same thing.

- Corrected.

L276-279: This part is not very clear. Do you select sigma y and sigma z separately? Could you end up with a sigma y of a given Pasquill-Gifford stability class and combine it with a sigma z from another stability class?

- We first determined sigma y based on the width of the plume observed during the measurement and the source location. From the distance between the source location and the maximum of the plume location and sigma_y we chose the most suitable Pasquill-Gifford stability class and then we chose the corresponding sigma_z value from the respective Pasquill-Gifford stability tables.

L288: It would be appropriate to at least in a sentence or two explain the isotopic analysis so the reader doesn't need to go back and read these papers (which analyzer, how long were the samples measured...).

Following this comment, we have now added and combined all information related to the isotopic analysis in Sect. 2.3.2 including analyzers, measurement time scales, calibrations, etc.

L314: Typo? "Utrecht and Hamburg correspond to" rather than "Utrecht and Hamburg were correspond to"
- Done

L321: Typo? "Figure 2" looks weird.
- Done

L332: You showed previously that different types of road had very different LI rates per km depending on cities, why didn't the authors use these road-specific emission factors to upscale their emissions?
- The evaluation showed that different types of road have different LI rates per km in these two cities, which means that the smaller or bigger LIs can happen on different road types. In this study we aim to compare cities based on total emissions derived from LIs, so for the upscaling we used total length of road no matter what road types those are.

Figure 5: Typo? "of collected air samples" instead of "of air samples collected". The authors should also show the microbial and pyrogenic clusters on these figures (L342).
- Done
- We tried to add additional boxes to the figure, but this makes the figure quite busy and therefore we prefer to highlight the "gas leaks" category only.

L352: Typo? "combustion-related" instead of "combustion, related".
- Done

L360: Not clear which criteria for $CH_4/CO_2$ ratios the authors used to classify LIs as combustion-related in the end. $CH_4/CO_2 > 0.2$ ppb/ppm?
- The criteria we used to identify combustion-related signals are based on $CH_4$ enhancement to $CO_2$ enhancement ($CH_4:CO_2$ ratio (ppb:ppm)). If the ratio is between 0.02 and 20 ppb:ppm and linear regression enhancements of these two species has $R^2$ greater than 0.8, we attributed those LIs as combustion-related sources. This has been specified in the revised version as follows (see Sect. 3.2, L469): *Based on the $CH_4:CO_2$ ratio (ppb:ppm) criterion defined above (see Sect. 2.3.1),…*

Figure 6: This figure is relatively difficult to interpret, it is difficult to visualize the shape of the observed plumes when they superimposed like this. It would have been interesting to see how and where you triangulated the location of the source for this site. How many sources did you find for this site? In wastewater treatment plant, the main methane source usually correspond to the sludge treatment areas that can be spotted with Google Earth.
- Figure 6 gives the overview of measurements around the WWTP. An example of the shape of a plume is given in Figure 7.

L375-377: The definition of the error estimate is very confusing, what are the 5 sets of measurements if there were only 3 days of measurements at the wastewater treatment plant?
- On some days there were two sets of measurements per day; e.g. one in the morning and one set in the afternoon. We have now defined the definition of measurement set in the paper which described back to back measurement downwind each facility as follows (see Sect. 2.2.3, L344-345): *A set of plumes is defined as a back to back transects during a period of time downwind each facility on different days.*

L395: Typo? Extra space before "74%".
- Done

L426: Typo? One of the "%" should be removed.
- Done

L413-432: The author should expand the discussion about the different source attribution approaches, is it necessary to use all of them? Which approach would the authors recommend to use in the future?
- The following sentence was added to the manuscript (see Sect. 4.2, L547-549): *Overall, $C_2H_6$ and $CO_2$ signals are very useful in eliminating non-fossil LIs in mobile urban measurements and with improvements in instrumentations, analyzing signals of these two species along with evaluation of $CH_4$ signals can make process of detecting pipeline leaks from NGDN more efficient.*

L479: Shouldn't it be the "annual natural gas leakage rate per capita" rather than the "annual natural gas consumptions per capita"?
- No, this sentence refers to the annual gas consumption provided in the previous mentioned sentence and intends to give a comparison between consumption per capita in Utrecht, Hamburg, and US.

L480: Typo? "per km of pipeline" rather than "per km pipeline"?
- Done

L491: The authors already explained several times that natural gas emissions depends on the age of the pipelines and the type of material used for these pipelines. I am not sure it is useful to repeat it here, especially since it will be discussed again later (L514).
- Here (Sect. 4.3), we mention the pipeline material and age, as these have important influences on the emissions from NGDNs in different cities, and later we give more information on different types and age of pipeline (see Sect. 4.4).

L545-549: The authors should choose one unit for the emissions and use it for all the sources, it would make easier for the reader to compare these emissions (wastewater treatment plant in t/yr, wells in kg/h...).
- Done

L557: Typo? "For emissions from the NGDN, the urban..." rather than "For emissions from the NGDN the urban...".
- Done

L545-557: Did the authors also looked at the ratios of these larger facilities? It could be also be an interesting information.
- Correlations between $CH_4:CO_2$ for the facilities were not very good, which may be due to the relatively small enhancements of $CH_4$ downwind the facilities and the expected ratio of $CH_4:CO_2$.
- For the Utrecht WWTP, a ratio of 0.4 ppb:ppm of $CH_4:CO_2$ with $R^2$ of about 0.52 was observed. The sludge treatment part of the WWTP emits both $CH_4$ and $CO_2$ while $CO_2$ is also emitted from other parts of the WWTP, e.g. power generation, anoxic/anaerobic treatment part, which explains why the correlation is not very high.

- Downwind the Compost and Soil Company in Hamburg the $CH_4$ enhancement was low and no clear correlation between $CH_4$ and $CO_2$ was observed.

Supplementary information: Section 1: "Figure S2a and Figure S2b show total length..." rather than "In Figure S2a and Figure S2b total length...are shown". Same for "In Table S1 and Table S2".
- Corrected

Section 2.1: Typo? Should it be "$CH_4$-only mode, which show" (rather than "$CH_4$-only mode. which show"). It is indeed very strange that the higher inlet measures higher methane enhancements than the bumper inlet. Would it possible that this source was located above the ground ("chimney" emissions or like the sewer pictures showed below)?
- The typo has been corrected, thanks for spotting this.
- Based on the $CO_2$ and C:C1 analysis this LI can only be attributed to a source of natural gas emission, likely from a pipeline leak in the ground.
- We are presently investigating the influence of intake height and instrument response in more detail for an upcoming publication, where measurements in several cities will be compared. Qualitatively, the relatively slow flush time of the cavity and lower measurement rate in the G2301 relative to the G4302 instrument (see comment above) lead to generally higher maximum enhancements in the G4302 instrument compared to the G2301, which for our measurements in Hamburg and Utrecht counteracts the fact that the inlet of the G2301 is closer to the ground and thus closer to most emission points. For individual plumes, turbulence in the street from driving cars can occasionally lead to higher mole fractions at the top inlet.

Section 2.2: What does "the ratio of the sum of $CH_4$ enhancements (in ppb) to the sum of $CO_2$ enhancements (in ppm)" mean? Does it correspond to the area under the plume? There is no mention of Figure S7 in the text.
- The respective sentence was changed as follow (see Sect. S.2.6): *In Figure 12, the ratio of the area under the $CH_4$ enhancements along the driving track (in ppb\*m) to the area of $CO_2$ enhancements along the driving track (in ppm\*m) is 5.5 ppb:ppm which is much higher than reported in previous studies, possibly indicating incomplete combustion.*

Section 2.4: "Errors in wind speed are estimated to be ±10% and for wind direction ± 5°" this seems low to me considering that the wind was not measured on site but at a tower located away from the site. Table 5 caption should be better isolated from Table 4, this is a bit confusing at the moment.
- By comparing some of the recorded measurement from the 2-D anemometer next to the facilities with the data from the towers we noticed that the local wind speed data were within ± 10% of the data from the towers; as described above, following the comment of the referee we now use a more conservative estimate of ± 30%.
- Given that for many sources the emission point is known, either from satellite imagery or from triangulation, wind direction (between emission point and maximum of observed plume is quite well known and here we think that the error estimate of ± 5° is adequate.
- The caption for Table 5 has also been corrected.

Section 2.6: In Figure S10a, shouldn't the authors constrain delta[13]C, deltaD, $C_2H_6$ and $CO_2$ before clustering? It would avoid clustering enhancements from different types of sources. Figure S11: caption not very precise.

- Based on the Weller et al. (2019) algorithm, it is assumed that LIs which are clustered together should be from the same source. Thus, based on the algorithm if one of the LIs within a cluster belongs to a specific emission class (e.g. microbial or combustion, etc.) then all the others should have fall into that source class. Based on the multi-tracer and isotope data, we have no evidence that in our dataset this is not the case. Therefore, we kept the analysis this way to keep consistency with other studies where no attribution techniques were used to attribute the LIs.

**References**

von Fischer, J. C., Cooley, D., Chamberlain, S., Gaylord, A., Griebenow, C. J., Hamburg, S. P., Salo, J., Schumacher, R., Theobald, D. and Ham, J.: Rapid, Vehicle-Based Identification of Location and Magnitude of Urban Natural Gas Pipeline Leaks, Environ. Sci. Technol., 51(7), 4091–4099, doi:10.1021/acs.est.6b06095, 2017.

Turner, D. B.: Workbook of Atmospheric Dispersion Estimates, U.S. Environmental Protection Agency. [online] Available from: https://nepis.epa.gov/Exe/ZyPDF.cgi/9101GKEZ.PDF?Dockey=9101GKEZ.PDF, 1969.

Weller, Z. D., Yang, D. K. and von Fischer, J. C.: An open source algorithm to detect natural gas leaks from mobile methane survey data, edited by M. Mauder, PLoS One, 14(2), e0212287, doi:10.1371/journal.pone.0212287, 2019.

Weller, Z. D., Roscioli, J. R., Daube, W. C., Lamb, B. K., Ferrara, T. W., Brewer, P. E. and von Fischer, J. C.: Vehicle-Based Methane Surveys for Finding Natural Gas Leaks and Estimating Their Size: Validation and Uncertainty, Environ. Sci. Technol., acs.est.8b03135, doi:10.1021/acs.est.8b03135, 2018.

---

## Author Comment (AC3) · 10 Nov 2020

This document provides our answers to the anonymous referee #2 for "**Methane mapping, emission quantification, and attribution in two European cities; Utrecht, NL and Hamburg, DE**"

We would like to extend our appreciation to the anonymous referee for the insightful and point-by-point comments from careful reading of our paper. In the following sections, comments from the referee are provided in normal black text, our replies are in normal blue, and changes is the manuscript are in *blue italic* format.

**1 Overview**

The authors present an extensive study of ground-based mobile measurements of methane and several related tracers ($C_2H_6$, $CH_4/CO_2$ ratio, $\delta^2H$-$CH_4$ and $\delta^{13}C$-$CH_4$ focused on quantifying and attributing methane emissions in two European cities, namely Utrecht and Hamburg, which both rely on subterranean pipelines as the delivery system for natural gas used in the households and otherwise. Such delivery systems are known to cause leaks that contribute to the anthropogenic global warming, and it has been demonstrated previously that fixing of these gas leaks can be a very cost-effective mean of climate change mitigation. Using a combination of in situ observations (with CRDS), discrete samples collected for identified leak sources as well as Gaussian plume modelling, the authors are able to identify approximately 100 leak sources in both cities over their study period, and thanks to the robust analysis of collected data, are able to differentiate them according to emission source (natural gas distribution system / microbial sources) and the respective source strength (with just several sources responsible for large parts of total emissions). A comparison of the results against previous studies conducted in US point to potential lower specific emissions for the studied cities. The authors also attempt to upscale the measurements performed over these limited campaigns in order to compare them to the publicly available aggregated data, albeit these results should be treated with care as the dataset is limited and much more robust studies are needed to achieve this goal (which the authors accurately point out).

The authors should be commended for the impressive amount of high-quality work that was put into design, execution and data analyses during those campaigns. This is no easy task, as the study encompassed simultaneously using many state-of-the-art techniques from very different scientific fields together with a very large amount of data (both measurement and supplementary) in order to achieved the stated goals. Simultaneous analysis of several tracers and isotopic composition is of particular interest, as it shows great promise in development the methods of precise small-scale emission estimations. I find this study to be a strong contribution to the discussion in the city-scale methane emissions, and the strategy developed here seems to be promising in developing both research-targeted and operational methods for leak detection and its strength estimation.

The article does suffer however, from this wealth of data and methods, and requires multiple improvements before final publication. For example, some sections of the text require more detailed information in order for the study to be considered reproducible. Also, the treatment of uncertainty in the source estimation and Gaussian plume modelling sections should be deepened. In the second case, the sensitivity of the method to the chosen meteorological parameters should be established. In methods section some restructuring is recommended, and the Discussion could benefit from introducing a clearly defined structure in order to appropriately focus attention.

I recommend publishing the article after addressing items listed below.

**2 Major comments:**

1. The Method description could do with an overhaul. In some places, more details need to be provided (see 'detailed comments' section below) for the experiments to be considered reproducible. In others, some information should be combined (2.6.4. and 2.4.).
 - We have revised the method section accordingly, see our answers to the detailed suggestions below.

2. Discussion of uncertainties in the urban emissions is very limited, with authors stating that 'We used a Bootstrap method (Nelson, 2008) to estimate 247 emission uncertainties similar to Weller et al. (2018) for the US city studies by resampling from all recorded LIs randomly 30,000 times.' No further comment is given, and in the discussion section the authors quickly skim over this and analyze the statistics of LI, without providing information on how precise those classifications might be. The method described by Weller et al. (and earlier by von Fisher et al.) relies heavily on assumptions regarding the distance to the source and (calibration using control releases) it stands to reason that this simplistic approach must produce very large uncertainties if not supported by multiple measurement repetition. This is critically important here as the data from limited detections is interpreted and up-scaled. As it is now, it is not possible to get a realistic impression about the numbers given, and puts the resulting data analysis in question. The discussion of uncertainty and potential biases should therefore be expanded.
 - As written in the manuscript, we follow the algorithm that has been described in detail in von Fischer et al. (2017) and Weller et al (2019). These studies include information on the precision, and we do not think that it is necessary to repeat all of the work that was performed there in our manuscript. We follow their approach so closely because we feel the inter-comparison with the US studies is important for a better overall understanding. The algorithm was applied to LIs that were surveyed at least twice, which we did not have as a strict criterion in our study. This difference is discussed in detail in our manuscript. Revisiting is not expected to have a large influence on the classification, since the Weller et al. (2019) algorithm only considers the maximum reading of the (at least two) identifications, so a revisit can only result in a higher emission rate estimate.
 - Like other studies, we do not have information on the distance of the leaks from the point of observation, since we do not have access to the full grid of pipelines in both cities. We do mention, that individual leak rate estimates can have large errors, and have included this statement explicitly regarding the highest derived leak rates in the revised manuscript. Since we covered a very large fraction of the city, the error of the total emissions is relatively small, i.e., about 30 % and 15 % in Utrecht and Hamburg, respectively. The bootstrapping is a standard technique and it has been demonstrated that it performs well in emission quantification and the determination of emission factors. The bootstrapping is a standard technique and it has been proven that it performs well in emission quantification and the determination of emission factors.
 - The following paragraph is now expanded in Sect. 2.2.2, L292-301:
 *To account for the emission uncertainty, similar to Weller et al. (2018) for the US city studies, we used a bootstrap technique which was initially introduced in Efron (1979, 1982), as this technique is adequate in resampling of both parametric and non-parametric problems with even non-normal distribution of observed data. Tong et al. (2012) indicated that bootstrap resampling technique is sufficiently capable in estimating uncertainty of emissions with sample size of equal or larger than 9. Efron and Tibshirani (1993) suggested that minimum of 1,000 iterations are adequate in*

*bootstrap technique. In this study, we used non-parametric bootstrap technique to account for the uncertainty of total CH$_4$ emissions from all LIs in each city with 30,000 replications. As mentioned above the algorithm is based on CH$_4$ enhancements of measurement with 5-40 m distance from controlled release location, and can produce large uncertainty for emission quantification of individual LI (Figure 4 in Weller et al. (2019)), but with sufficient number of sample size, the uncertainty associated with total emission quantified in an urban area is more precise.*

3. The authors state that 'Emissions from facilities show significant contributions to the total emissions in both cities.' (supplement), but many details on the method used for estimating them are missing. This section needs to be expanded and more info should be given about the analysis as well as the uncertainty estimation and sensitivity of the method to the stated assumptions. For example, the use of measurement data from distant towers in order to drive the transport model raises an eyebrow, as these are critical for calculating the emission rate. What was the average distance between the tower and the measurement location and what was the elevation of that measurement? Are the wind speed and direction uncertainties reasonable? What about the elevation of the source, which is only very briefly discussed in the supplement? In the end, the reader should have a comprehensive view on whether the method is able to provide good emission estimation in a given setting, and at the moment the result with error bars (on the order of ±50 % 1-σ) suggest it is not. This should also be discussed in more detail.

- The focus of our paper is on the detection of methane elevations from unknown sources with the mobile vehicle approach and their attribution to natural gas, microbial and combustion sources. In addition to this new attribution component, we also decided to report rough emission rates derived for larger known facilities, in contrast to previous publications that excluded emissions from such facilities. The reason was to demonstrate that such elevations are also picked up by the mobile measurements, to estimate whether estimates on emission rates can be derived, even if the approach was not targeted to facilities, and to put the order of magnitude of emissions from the unknown sources into perspective of the known sources (e.g. waste water treatment plants).

We kept this section short on purpose to not distract too much from the main objective but realize from the referee reports that a more detailed is warranted. Some more description has been incorporated in the revised version of the manuscript, and the replies to the specific questions are included in our answers to the detailed comments below.

In addition to the information which had been already provided in supplementary section we now have expanded the paragraphs related to the emission height (L333-341) and wind speed (L351-358) in Sect. 2.2.3 as follows:

*Neumann and Halbritter (1980) showed that the main parameters in sensitivity analysis of GPDM are the wind speed and source emission height in close distance and the influence of emission height become less further downwind compared to the mixing layer height. In this study, the heights of emission sources were low (<10m) and/or estimated during surveys and Google Earth imageries, and considering that such a larger measurement distance from the facilities, the main sources of uncertainty of the emission estimates for the WWTP and Compost and Soil company are most likely the mean wind speed and for the upstream facilities in Hamburg the major sources of uncertainties can be the mean wind speed and emission height. We considered 0-4 m source height for the WWTP in Utrecht, and for the upstream facilities in Hamburg we considered 0-5 m emission height for the Compost and Soil site, 0-2 m for the separator,*

*0-10 m for the storage tank, and 0-1 m for the oil extraction well-head. We used 1 m interval for each of these height ranges to quantify emissions in GPDM.*

*Due to technical issues, local wind data were not logged continuously and thus we used wind data from two towers which are 8 to 20 km away from the facilities we focused for emission quantifications. These distances introduce extra uncertainties in analyzing the emissions using GPDM mainly on the wind speed. By comparing some of the local high-quality wind data to data from the towers, we estimated that the local wind speed is within the range of ± 30 % of the collected tower data. This range was adopted to estimate the wind speed for emission quantifications for the set of plumes measured downwind of the facilities. The wind directions were aligned at local scale of each facility based on the locations of sources and locations of maxima of average $CH_4$ enhancements from a set of transects in each day's survey and we considered ± 5° uncertainty in wind direction for the GPDM quantification.*

4. The discussion section would benefit from introducing subsections to provide focus for specific items under discussion.

- This has been updated, and we agree that this provides a clearer structure.

5. I find the overall quality of language very high, yet there are multiple minor deficiencies that still need to be addressed. Below I have listed some of them. I believe that this is mostly due to heavy editing during manuscript preparation, and I ask the authors to take special care of that issue before resubmitting.

- Thanks for pointing out several detailed issues, which have all been incorporated (see detailed replies below). The manuscript has also been carefully proof-read again.

6. The supplement is large and – I'm sorry to say that – poorly edited (tables are too large - the font can safely be made much smaller; order of figures and sections does not correspond to the manuscript reference order). In some cases, itis a source of important information that is also in some places missing from the main text (already mentioned section 2.4. and corresponding S.2.4, figure S16). If the authors want to keep some technical details apart from the main text (understandable with that much material), then I would ask to consider putting the more important parts in the Annex, in order to a) maintain the high editing standard and thus make reading easier, b) keep the important information together with the text. At the very minimum the editing of the supplement needs to be improved.

- The editing of the supplement has been updated according to the suggestions of the referee. We have moved some of the text from the supplement to the main text, as suggested.

**3 Detailed comments:**

Line numbers are given for identification. Comments for figures are given at the end of the list.

L25, also later in the text: ACP requires exponential notation of units, consult the 'manuscript preparation' on the ACP website for details (www.atmospheric-chemistry-and-physics.net/for_authors/manuscript_preparation.html)

- This has been adjusted in the revised version of the manuscript

L35: I'd suggest putting ppm outside of parentheses and mole fraction inside, as the ppm/ppb notation is the dominant one in the manuscript.

- The sentence was changed accordingly (L35-36): *The increase of CH₄ mole fraction from about 0.7 parts per million (ppm) or 700 parts per billion (ppb) ...*

L46-49: Sentence needs rephrasing
- The sentence was rephrased as follows (see Sect. 1, L47-50): *CH₄ emissions originate from a wide variety of natural and anthropogenic sources, for example emissions from natural wetlands, agriculture (e.g. ruminants or rice agriculture), waste decomposition, or emissions (intended and non-intended) from oil and gas activities that are associated with production, transport, processing, distribution, and end-use of fossil fuel sector (Heilig, 1994).*

L60-62: This paragraph does not fit well here, would be better if info given as part of previous or next.
- The information has been added to the end of previous paragraph (L60-63).

L64: 'high precision' used twice
- Corrected

L75-77: What were the main findings from these studies? Specifically, it would be good to comment on whether these methods can be useful for up-scaling.
- The following information has been added to the manuscript (see Sect. 1, L78-86): *Gioli et al. (2012) showed that about 85 % of methane emissions in Florence, Italy originated from natural gas leaks. Helfter et al. (2016) estimated CH₄ emissions of 72 ± 3 t km⁻² yr⁻¹ in London, UK mainly from sewer sesytem and NGDNs leaks, which is twice as much as reported in the London Atmospheric Emissions Inventory. O'Shea et al. (2014) also showed that CH₄ emissions in greater London is about 3.4 times larger than the report from UK National Atmospheric Emission Inventory. Zimnoch et al. (2019) estimated CH₄ emissions of (6.2 ± 0.4) × 10⁶ m³ year⁻¹ for Krawko, Poland, based on data for the period of 2005 to 2008 and concluded that leaks from NGDNs are the main emission source in Krawko, based on carbon isotopic signature of CH₄. Chen et al. (2020) also showed that incomplete combustion or loss from temporarily installed natural gas appliances during big festivals can be the major source of CH₄ emissions from such events, while these emissions have not been included in inventory reports for urban emissions.*

L78: 'We quantified emissions in this study using measured CH₄ enhancements above background, which were detected' - This needs revision; also, it feels like Weller et al. 2019 should already be quoted here, perhaps something like: 'In this study, we have quantified the CH₄ emissions using the method described by Weller et al., who demonstrated...'
- The following sentence has been added to the manuscript (see Sect. 1, L88-90): *In this study, we quantified LIs emissions using an empirical equation from Weller et al. (2019), which was designed based on controlled release experiments from von Fischer et al. (2017), to quantify ground-level emissions locations in urban area such as leaks from NGDN.*

L91: Was the reproducibility tested by the authors? Picarro currently gives 0.5 ppb for 5 s raw data. If the reproducibility was tested by the authors, please provide some details on the testing (either reference, or brief description of the experiment). Was the water correction modified or the factory settings were used? Please state that explicitly and also provide information if necessary.

- We now added more info about the G2301 to the manuscript (see Sect. 2.1.1, L100-108).
- The numbers represent the approximate range of the instrument noise when measuring background air. This information is provided to indicate to which order of magnitude we can identify elevations of $CH_4$ and $C_2H_6$.
- The factory settings were used for the water correction.

L99: Discard 'about' or the approximation sign
- Done

L100-101: Similar to comment for L91, please provide more info.
- Similar to the details provided for G230, information about the Picarro G4302 is now expanded. We also used factory setting to consider water correction (see Sect. 2.1.1, L109-117).

L104: Info on how the delay was calculated should be given here, but can be found later in L202. Please combine both (see major comment no. 1)
- Done

L111-112: Please spell out the main findings of the discussed comparison. Also, the reference to annex section number (S.2.) where it is discussed should be present (next to table S3 ref.). In general, sections of supplement should be referenced and not only tables or figures from it.
- The comparison sentence is now edited as follows with the main findings from the comparison (see Sect. 2.1.1, L135-142): *A comparison between the two instruments during simultaneous measurements showed that all LIs were detected by both instruments despite difference in instrument characteristics and inlet height. In the majority of cases $CH_4$ enhancements for each LI from both instruments were similar to each other. We note that there is likely a compensation of differences from two opposing effects between the two measurement systems. The inlet of the G2301 was at the bumper, thus closer to the surface sources, but the rather low flow rate and measurement rate of the instrument lead to some smoothing of the signal in the cavity. Because of the high gas flow rate, signal smoothing is much reduced for the G4302, but the inlet was on top of the car, thus further away from the surface sources (see* **Error! Reference source not found.** *in SI, Sect. S.1.3).*

L129: When reading the sentence for the first time I have understood that the gas pipeline network corresponds to the street network 1:1. Is that correct, or the general coverage of municipal areas is meant? Please clarify.
- That is correct. The sentence was changed as follows (see Sect. 2.1.2, L162-165): *The local distribution companies (LDCs) in Utrecht (STEDIN (https://www.stedin.net/)) and Hamburg (GasNetz Hamburg (https://www.gasnetz-hamburg.de)) confirmed that full pipeline coverages are available beneath all streets. Therefore, the length of roads in the study area of Utrecht and Hamburg are representatives of NGDNs length.*

L137: discard 'at the following links: Utrecht and Hamburg'
- Done

L147-148: Please briefly explain how the vehicles can be methane sources (with reference for subsequent discussion further in the text).
- The sentence was changed as follows (see Sect. 2.1.3, L183-185): *For example, they could be related to emissions from vehicles which run on compressed natural gas, or*

*vehicles operated with traditional fuels but with faulty catalytic converter systems. Later we will discuss how to exclude or categorize these unintended signals (see Sect. 2.2.2 and Sect. 2.3.1).*

L148: Please state clearly how many revisits were usually made.
- The respective sentence was changed as follows (see Sect. 2.1.3, L185-187): *Therefore, we revisited a large number of locations (65 in Utrecht (≈80 %) and 100 in Hamburg (≈70 %)) where enhanced $CH_4$ had been observed in during the first survey in order to confirm the LIs.*

L159-161: Have any cases where new leaks have occurred in-between surveys been observed?
- There have been some locations where we observed new LIs during revisits but not in the earlier visits or all the way round. This can be mainly due to the fact that not all LIs (mostly small LIs) are observable in all visits.

L166-167: Parentheses missing? 3 L bag for a price of 2 L bag is too good to be true.
- Corrected

L168-169: More details on sampling are needed.

- Was the data collection stationary or also mobile?
  The following sentence has been added to the manuscript (Sect. 2.1.4, L206-209):
  *All the samples taken in the North Elbe study area and from most of the facilities were collected when the car was parked, but the samples inside the New Elbe tunnel and close to some facilities where there was no possibility to park were taken in motion while we were within the plume.*

- How was the plume / non-plume location determined?
  The following sentences have been added to the manuscript (Sect. 2.1.4, L209-211):
  *The sampling locations across the North Elbe study area of Hamburg were determined based the untargeted surveys, and the confirmation during revisits. The $C_2H_6$ information was not used in the selection of sampling locations in order to avoid biased sampling.*

- What was the flushing time?
  The samples were taken using a pump with flow rate of 0.25 L min$^{-1}$. [info added to the sampling section (Sect. 2.1.4, L214)].

- Was the sample dried? How?
  Samples were dried in the lab followed by the $CH_4$ extraction. [info added to the lab analysis section (Sect. 2.3.2, L383-384 and L391-392)].

L173: See major comment no. 3.
- The relevant section has been expanded to provide more information (see Sect. 2.2.3). The reason for using data from the two mentioned towers in Utrecht and Hamburg is that the online data logging setup failed to continuously record all the local wind measurements during the surveys. The distance of the towers to the facilities ranges from 8 to 20 km, and indeed these distances introduce extra uncertainties in emission quantification mainly wind speed.

- When we compare the data that were recorded on the vehicle with the tower, we derive a difference in wind speed of ± 10 %. After considering the remarks of the referee, we increase this to a more conservative error estimate of ± 30 %. Regarding the wind direction (e.g. the oil wells), for several sources the emission point is relatively certain and can by analysis of Google Earth images. In addition, we passed several sources during different wind conditions and did a "triangulation" based on the observed plumes and wind data (see Sect. 2.2.3, L352-359).

L191-192: Case shown in S5 is special and I strongly recommend to remove it from here and discuss later. As it is now, the text does not explain it, and thus may imply that all the cars are potential sources of $CH_4$, which is certainly not the case.
- Section 2 has been rearranged and this part has been moved to the sub-section of '2.3. Emission attributions' (Sect. 2.3.1, L367-370). The text has been revised as follows to mention that not all vehicles emit $CH_4$ but vehicles running on compressed natural gas. *Compressed natural gas vehicles can be mobile $CH_4$ emission sources ( E. K. Nam et al., 2004; Curran et al., 2014; Naus et al., 2018; Popa et al., 2014) and in this study we also observed $CH_4$ signals from vehicles. For example, the point to point $C_2H_6$:$CH_4$ ratio ($C_2$:$C_1$) calculated from road measurements of a car exhaust shown in* **Error! Reference source not found.** *is 14.2 ± 7.1 %.*

L194: Consider providing these standards in the parentheses or the supplement section S.7.
- Done

L196: How many such cases were observed? Could they be important for the overall budget?
- In the study are of Hamburg (north Elbe), there were 34 cases and in the study area of Utrecht (inside the ring) there were 7 cases which we excluded based on this constraint (C2:C1 >10). Note that the methane/ethane instrument (Picarro G4302) was used for all surveys in Hamburg, while this instrument was not available for all surveys in Utrecht.
- The $CH_4$ enhancements measured by the G4302 for these cases were 0.37 ± 0.24 ppm in Hamburg and 0.26 ± 0.03 ppm in Utrecht. Based on the quantification from Weller et al. (2019), these LIs are not important to the overall budget but they will of course affect the total number of leak indications (LIs). Therefore, it is important to exclude them from the evaluation.

L198: Please state the reasons for this exclusion, briefly.
- The speed constraints are now reformulated as follows (see Sect. 2.2.2, L253-256): *Following the algorithm from von Fischer et al. (2017), measurements at speeds above 70 km $h^{-1}$ were excluded, as the data from the controlled release experiments (von Fischer et al., 2017) were not reliable at high speed (Weller et al., 2019). We also excluded measurements during periods of zero speed (stationary vehicle) to avoid unintended signals coming from other cars when the measurement car was stopped in traffic.*

L199-200: Just a small comment, no action needed: I don't see the benefit of this artificial increase of the data frequency. This brings no new information at the cost of tripling of the data that needs to be processed.

L200-201: Wording. If the time was just converted to UTC, then calling it 'a correction' is not warranted. Consider changing to: 'Following the interpolation step, the data was converted to UTC, and subsequently corrected for ...'

- We used the word 'correction' as the clock on the Picarro instruments had a drift over the period of the campaign and we needed to correct this drift to set all the measurements to the correct UTC time. The text has been adjusted as follows:
  The clocks on the Picarro instruments were set to UTC but showed drift over the period of the campaigns. We recorded the drifts for each day's survey and corrected to UTC time. The data were also corrected for the delay between air at the inlet and the signal in the $CH_4$ analyzers. This delay was determined by exposing the inlet to three small $CH_4$ pulses from exhaled breath, ranging from 5-30 seconds, depending on the instrument and tubing length. We averaged the three attempts to determine the delay for each instrument and used the delays for each instrument. Individual attempts were 1 to 2 s different from each other. For the G4302 the delay was generally about 5 s and for the G2301 it was about 30 s; the difference is mainly due to the different flow rates. The recorded $CH_4$ mole fractions were projected back along the driving track according to this delay.

L202: About the delay time estimation: 5–30 seconds is a very broad range. Were the ranges so variable for both instruments, or was it 5 for one and 30 for the other? Also, how was the pulse generated? Can you estimate precision of that delay estimation (even grossly)?

- The answers to this comment are merged with the answers to the previous comment (see Sect. 2.1.1, L122-129).

L204: Reference order needs correction. Previous reference supplement figure was S5 (L190).

- Corrected

L207: In $CO_2$ signal (Fig. S8), it can be clearly seen that the background line is some-times higher than the observed signal. Since this plot is about the background, it would be good to change the limits of y axis to make the calculated background visible clearer, especially for methane. Please give some comment about the possible negative enhancements after subtracting such background (can it affect the estimation of emissions?).

- For $CO_2$, individual plumes, e.g. from vehicles, can overlap and create a locally enhanced background according to our background extraction procedure (5th percentile of $CO_2$ measurements in a ± 2.5 min time window). Negative deviations are not considered, but the enhanced background may result in a potential small underestimate of the $CO_2$ enhancement. Such small changes will not affect the categorization of the $CH_4$ enhancement based on the $CH_4:CO_2$ ratio, very wide ranges were assumed for this source attribution. We have updated the figure in the revised version as suggested (see Figure S7 in SI, Sec. S.2.1).

L217: Please add 'peak' after enhancement, to make it clear that it's not about the release height.

- We changed this to: *...to demonstrate that the magnitude of the observed methane enhancement...* (see Sect. 2.2.2, L264)

L237: Wording. Why should results from different cities be comparable? The authors clearly mean that the analysis software used on a given dataset should be comparable. Please clarify. Actually, this whole paragraph can be limited to information that 'Our software was compared

to analysis tools developed by CSU (von Fisher et al.2017, Weller et al. 2019) and no significant differences were observed (see SI, section S.2.7)'.

- Thank you for the suggestion, we updated the paragraph as follows (see Sect. 2.2.2, L284-286): *We compared the outputs of our software to the one developed by Colorado State University (CSU) for the surveys in US cities (von Fischer et al., 2017; Weller et al., 2019). 30 LIs were detected and no significant differences were observed (linear fit equation y = 1.00 \* x - 0.00, R² = 0.99) (see SI, Sect. S.2.4, Figure S10).*

L251: Erase 'areas'.
- Done

L252: Erase empty parentheses.
- Done

L253 and L384: 'drive-by'- I propose 'mobile'. I was surprised to find it used in Fisher et al. (albeit only once), as in U.S. this word is sometimes used to describe something much more nefarious then GHG observations.
- Done

L254: 'We report (...)' – Unclear what is meant in this sentence, please rephrase it.
- The sentence was rephrased as follows (Sect. 2.2.3, L309-310): *In this study, we also report the data obtained from larger facilities, since rough emission estimates from facilities can be obtained in the city surveys.*

L256-257: Erase 'both' and 'each day's'
- Done

L269-274: How was the release height determined? How is the uncertainty of this determination included in the uncertainty of emission?
- Information on the release height and uncertainty associated with each facility has been added to this section (see answer to the major comment 3).

L283: This section should be combined with 2.4.
- Section 2 has been rearranged and information regarding air sampling collection is now provided in Sect. 2.1.4 and details on analysis of samples are provided in Sect. 2.3.2.

L286: Info on the isotopic scales used in this study needs to be given.
- The information on the scale and instruments for isotopic measurement is now added (see Sect. 2.3.2, L380-383 and L393-395).

L293-294: Please provide explanation on why these particular ranges were selected. For signatures, specifically also provide references supporting the choice of isotopic signatures. Please keep in mind that for fossil fuel signatures, figure 7 from Rockmann et al. 2016 doesn't give a full picture – see e.g. Sherwood et al. 2017 for a broad overview of isotopic signatures for fossil methane.
- We acknowledge that the full range of isotopic signatures of natural gas from different reservoirs is wide, in particular for $\delta^{13}C$, as documented in Sherwood et al. (2017), and we have indeed adjusted this range in a recent publication of Menoud et al., (2020). Nevertheless, this does not mean that the full range of signatures would be encountered in one region in a single campaign. No data are available for Hamburg, but Levin et al., (1999) reported a mean $\delta^{13}C$ value of -40 ‰ for the NGDN in Heidelberg, with a

seasonal cycle of ± 10‰. Hoheisel et al. (2019) reported $\delta^{13}C$ = -43.3 ± 0.8‰ for the NGDN in Heidelberg the period 2016-2018, but without the strong seasonality, and -46.1 ± 0.8‰ for natural gas storage tanks and compressor stations. These values are compatible with the selected range that we have chosen based on Röckmann et al. (2016).

L314-315: 'were correspond' – corresponded
- Done

L315: What is the uncertainty here? This relates to major comment 2.
- See the answer to major comment 2. We have also added the following sentence (Sect. 3, L422-424): *Noted that estimates for individual leaks with the Weller et al. (2019) algorithm can have large error, thus these results are indicative of large leaks, but the precise emission strength is very uncertain.*

L378: Why uncertainty given only for wind direction?
- Corrected

L394: von Fisher – V should be capitalised at the start of the sentence (Von Fisher).
- Done

L404: 'About 50 %' - please give the specific number that was used in the calculation.
- We used precisely 50 % for both cities for the calculation. The wording has been adjusted.

L420-422: These numbers are in fact very similar. The variability of $\delta^{13}C$ in the natural gas can be quite substantial. See e.g. Fig 4 in Sherwood et al, 2017.
- We have updated this sentence as follows (see Sect. 4.2, L533-537): *These numbers do not agree within combined errors, but are also not very different. $\delta^{13}C$ values of $CH_4$ from the NGDN can vary regionally and temporally, e.g. due to differences in the mixture of natural gas from various suppliers for different regions in Germany (DVGW, 2013). It is also shown that how $\delta^{13}C$ values of fossil fuel $CH_4$ have significant variabilities in different regions at global scale (Figure 4 in Sherwood et al. (2017)).*

L426: % used twice
- Corrected

L433: Reference to Figure 1?
- Corrected

L436-438: a) Fig S16 also points to the local sewage system as potentially important source, but this is not mentioned here. b) Please be more precise in the argument here- i.e. explain why measurements around the lake point to anaerobic methanogenesis specifically. Linking to a), please include info on the potential role of the sewage system if needed. Is it possible that the sewage is seeping into the lake?
- a) We changed this sentence as follows (see Sect. 4.2, L553-559): *Many of the microbial LIs encountered in Hamburg are around the Binnenalster lake (**Error! Reference source not found.**), which suggests that anaerobic methanogenesis (Stephenson and Stickland, 1933; Thauer, 1998) can cause these microbial emission in this lake, as seen in other studies focused on emissions from other lakes (DelSontro et al., 2018; Townsend-Small et al., 2016). Microbial $CH_4$*

*emissions from sewage system (Guisasola et al., 2008) can also be an important source of in this area, as seen in US urban cities (Fries et al., 2018). Fries et al. (2018) performed direct measurement of $CH_4$ and nitrous oxide ($N_2O$) from a total of 104 sites, and analyzed $\delta^{13}C$ and $\delta D$ signatures of samples from 27 of these locations, and attributed 47 % of these locations to microbial emissions in Cincinnati, Ohio, USA.*

    b) We described in the text that the isotope and multi-tracer observations point to microbial sources. Unfortunately, we do not know if and/or to what degree the sewage system seeps into the lake.

L443: 'because there is no publicly available activity data for associated activity data'– please rephrase.
- The sentence was rephrased as follows (see Sect. 4.3, L564-566): *Also, it is not possible to calculate a robust city-level estimate using the nationally reported emission factors because there is no publicly available associated activity data, i.e., pipeline materials and lengths for each material, at the level of individual cities.*

L452: Too many parentheses. I suggest '(...) 40 kg km$^{-1}$ yr$^{-1}$ (for other material, p <200 mbar; see p. 130 in Peek et al., 2019, for details)
- The sentence was split in several parts and rewritten as follows (see Sect. 4.3, L571-577): *The Netherlands National Institute for Public Health and the Environment (RIVM) inventory report derived an average NGDN emission factor of $\approx$ 110 kg km$^{-1}$ yr$^{-1}$ using 65 leak measurements from different pipeline materials and pressures in 2013. This weighted average ranged from a maximum of 230 kg km$^{-1}$ yr$^{-1}$ for grey cast iron pipelines to a minimum of 40 kg km$^{-1}$ yr$^{-1}$ for pipelines of other materials with overpressures <= 200 mbar (for details, see P. 130 in Peek et al. (2019)). This results in an average $CH_4$ emissions of $\approx$ 70 t yr$^{-1}$ (min = 30 t yr$^{-1}$ and max = 150 t yr$^{-1}$) for the study area of Utrecht, assuming $\approx$ 650 km of pipelines inside the ring, and further assuming that Utrecht's NGDN is representative of the national reported average (see qualifiers above).*

L473: 'credibility interval' – confidence interval
- Done

L491: '(...) factors can be gas pipeline age and material, sewer system.' Part of sentence missing? Please rephrase.
- The sentence was changed as follows (see Sect. 4.3, L611-613): *$CH_4$ emissions can vary among different cities, depending on the age, management and material of NGDNs, and/or the management of local sewer systems.*

L531: 'were' – where
- Corrected

L533: 'as shown' used twice.
- Corrected

L542-544: The scheme from S18 cannot be treated as a 'protocol' without a proper description of the method. In reality, it describes the main components of the method applied in the study, so in fact the manuscript itself is more of such a protocol. As it stands now, consider either expanding the description in the supplement (so gas companies might actually use it as a protocol) or discard it altogether.

- We agree with this comment and removed the term "protocol". We reformulated the sentence as follows (see Sect. 4.4, L668-669): *Figure S19 (see SI, Sect. S.5) illustrates how the individual measurement components can be efficiently combined in a city leak survey program.*

L561: 'corresponding to emissions of about 107 tCH$_4$/ yr' – exactly 110 t yr$^{-1}$ is given in L332.
- Corrected

L562: Please state the method, e.g. 'These estimates, based only on the studied area, were then up-scaled for the total municipal area, using the road network map as a proxy to (...)'
- The up-scaled emissions are related to the emissions across the study area; inside the ring in Utrecht and north Elbe in Hamburg and not the total municipal area of these two cities. The text has been adjusted as follows (see Sect. 5, L688-690): *These estimates, based on the streets covered, were then up-scaled to the total study area, using the road network map as a proxy for the length of the pipeline network which then yielded total emissions of 150 t yr$^{-1}$ and 440 t yr$^{-1}$ across the study area of Utrecht and Hamburg respectively.*

L567: 'were from' - I suggest 'originated from'
- 'were from' has been changed to '*originated from*' (L695)

Figure 2: a) Please fix the x axis description - extra arrow unnecessary. b) the plot is cropped in the lower part, by several pixels. c) extra grid dashed lines (green) are unnecessary, make the labels difficult to read.
- a) We prefer to keep the arrow. It indicates that the total emissions are attributed to four different emission sources and how much these sources contribute to the total emissions individually
- b) Done, this probably happened during conversion to pdf format.
- c) The extra green dashed lines on the x-axis have been removed
   The caption has been changed as follows: *Total CH$_4$ emission rates from different sources in (a) Utrecht and (b) Hamburg; the arrow shows how the emissions are attributed to different sources*

Figure 3: a) again, the arrow from total emissions to 'Road classes' seems unnecessary. b) please explain the arrows from the plot in the caption.
   a) The arrow is used to indicate that the total emissions are categorized into six different road classes and how much these classes contribute to the total emissions individually
   b) the caption has been changed as follows: *Total CH$_4$ emissions in Utrecht and Hamburg; the arrow shows how the total emissions are distributed over different road classes*

Figure 4: I recommend plotting all US cities in a single colour (grey?) and simply label the line as ('15 US Cities', Weller et al.) or similar. The colours are indistinguishable anyway.
- Done

Figure 5: Excellent plot! a) Please add comment about the uncertainty of $\delta^{13}$C and $\delta^2$H signatures in the caption (they were only plotted for C2/C1. I also softly suggest to label the plots with a-b-c-d and move the labels into figure caption. b) On previous plot the units were placed in parentheses, consider keeping notation consistent.
- Both suggestions have been included

Figure 6: a) Plumes of what? b) Please also provide information on whether the peaks are on the same scale; if yes, then what is the plotted range of mole fractions (if those are mole fractions)?

- Done

**References**

DelSontro, T., Beaulieu, J. J. and Downing, J. A.: Greenhouse gas emissions from lakes and impoundments: Upscaling in the face of global change, Limnol. Oceanogr. Lett., 3(3), 64–75, doi:10.1002/lol2.10073, 2018.

von Fischer, J. C., Cooley, D., Chamberlain, S., Gaylord, A., Griebenow, C. J., Hamburg, S. P., Salo, J., Schumacher, R., Theobald, D. and Ham, J.: Rapid, Vehicle-Based Identification of Location and Magnitude of Urban Natural Gas Pipeline Leaks, Environ. Sci. Technol., 51(7), 4091–4099, doi:10.1021/acs.est.6b06095, 2017.

Hoheisel, A., Yeman, C., Dinger, F., Eckhardt, H. and Schmidt, M.: An improved method for mobile characterisation of δ 13 CH 4 source signatures and its application in Germany, Atmos. Meas. Tech., 12(2), 1123–1139, doi:10.5194/amt-12-1123-2019, 2019.

Levin, I., Glatzel-Mattheier, H., Marik, T., Cuntz, M., Schmidt, M. and Worthy, D. E.: Verification of German methane emission inventories and their recent changes based on atmospheric observations, J. Geophys. Res. Atmos., 104(D3), 3447–3456, doi:10.1029/1998JD100064, 1999.

Menoud, M., van der Veen, C., Scheeren, B., Chen, H., Szénási, B., Morales, R. P., Pison, I., Bousquet, P., Brunner, D. and Röckmann, T.: Characterisation of methane sources in Lutjewad, The Netherlands, using quasi-continuous isotopic composition measurements, Tellus B Chem. Phys. Meteorol., 72(1), 1–19, doi:10.1080/16000889.2020.1823733, 2020.

Peek, C. J., Montfoort, J. A., Dröge, R., Guis, B., Baas, K., Huet, B. van, Hunnik, O. R. van and Berghe, A. C. W. M. van den: Methodology report on the calculation of emissions to air from the sectors Energy, Industry and Waste, as used by the Dutch Pollutant Release and Transfer Register., 2019.

Sherwood, O. A., Schwietzke, S., Arling, V. A. and Etiope, G.: Global Inventory of Gas Geochemistry Data from Fossil Fuel, Microbial and Burning Sources, version 2017, Earth Syst. Sci. Data, 9(2), 639–656, doi:10.5194/essd-9-639-2017, 2017.

Townsend-Small, A., Disbennett, D., Fernandez, J. M., Ransohoff, R. W., Mackay, R. and Bourbonniere, R. A.: Quantifying emissions of methane derived from anaerobic organic matter respiration and natural gas extraction in Lake Erie, Limnol. Oceanogr., 61(S1), S356–S366, doi:10.1002/lno.10273, 2016.

Weller, Z. D., Yang, D. K. and von Fischer, J. C.: An open source algorithm to detect natural gas leaks from mobile methane survey data, edited by M. Mauder, PLoS One, 14(2), e0212287, doi:10.1371/journal.pone.0212287, 2019.

Weller, Z. D., Roscioli, J. R., Daube, W. C., Lamb, B. K., Ferrara, T. W., Brewer, P. E. and von Fischer, J. C.: Vehicle-Based Methane Surveys for Finding Natural Gas Leaks and Estimating Their Size: Validation and Uncertainty, Environ. Sci. Technol., acs.est.8b03135, doi:10.1021/acs.est.8b03135, 2018.